# Exploring Diffusion Transformer Designs via Grafting

**Keshigeyan Chandrasegaran** [*‡1,2]   **Michael Poli** [*1,2]   **Daniel Y. Fu** [3,4]   **Dongjun Kim** [1]
**Lea M. Hadzic** [1]   **Manling Li** [1,5]   **Agrim Gupta** [6]   **Stefano Massaroli** [2,7]
**Azalia Mirhoseini** [1]   **Juan Carlos Niebles** [1,8†]   **Stefano Ermon** [1†]   **Li Fei-Fei** [1†]

[1] Stanford University   [2] Liquid AI   [3] Together AI   [4] UC San Diego
[5] Northwestern University   [6] Google DeepMind   [7] RIKEN   [8] Salesforce Research

grafting.stanford.edu

## Abstract

Designing model architectures requires decisions such as selecting operators (e.g., attention, convolution) and configurations (e.g., depth, width). However, evaluating the impact of these decisions on model quality requires costly pretraining, limiting architectural investigation. Inspired by how new software is built on existing code, we ask: can new architecture designs be studied using pretrained models? To this end, we present *grafting*, a simple approach for editing pretrained diffusion transformers (DiTs) to materialize new architectures under small compute budgets. Informed by our analysis of activation behavior and attention locality, we construct a testbed based on the DiT-XL/2 design to study the impact of grafting on model quality. Using this testbed, we develop a family of hybrid designs via grafting: replacing softmax attention with gated convolution, local attention, and linear attention, and replacing MLPs with variable expansion ratio and convolutional variants. Notably, many hybrid designs achieve good quality (FID: 2.38–2.64 vs. 2.27 for DiT-XL/2) using $< 2\%$ pretraining compute. We then graft a text-to-image model (PixArt-$\Sigma$), achieving a $1.43\times$ speedup with less than a 2% drop in GenEval score. Finally, we present a case study that restructures DiT-XL/2 by converting every pair of sequential transformer blocks into parallel blocks via grafting. This reduces model depth by $2\times$ and yields better quality (FID: 2.77) than other models of comparable depth. Together, we show that new diffusion model designs can be explored by grafting pretrained DiTs, with edits ranging from operator replacement to architecture restructuring. Code and grafted models: grafting.stanford.edu.

## 1 Introduction

Model architecture design plays a central role in machine learning, alongside data, algorithms, compute, and benchmarks. It defines a learnable function and entails key decisions, including the choice of operators (e.g., attention, convolution) and configurations (e.g., model depth, width). Despite this, insight into architectures—what works and what doesn't—is difficult to obtain due to the prohibitive costs of training models from scratch, especially in today's foundation model era. As a result, studying new architectures remains a challenge, particularly for generative models. Much like how new software is built on existing code rather than written from scratch, can pretrained models serve as scaffolds for studying new architectures? In this work, we investigate *architectural editing of pretrained models to study new architecture designs.* We focus on diffusion transformers (DiTs), a class of generative transformers widely used for image and video generation [1, 2, 3].

---

[*] Equal contribution.   [†] Equal senior authorship.
[‡] Part of this work was done at Liquid AI.
Correspondence to {keshik,poli}@stanford.edu

39th Conference on Neural Information Processing Systems (NeurIPS 2025).

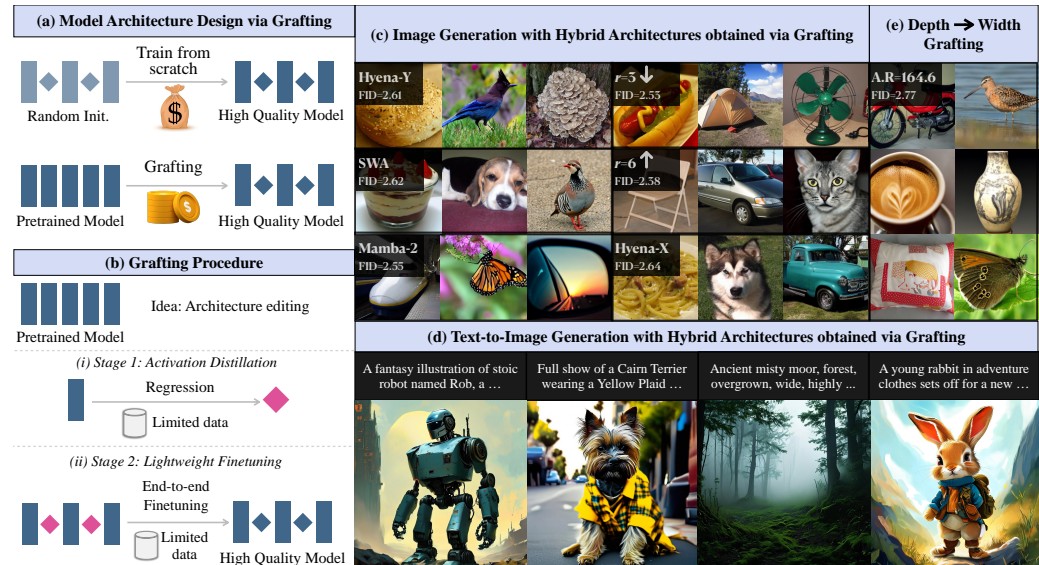

Figure 1: **Grafting overview.** *(a,b) Model architecture design via grafting.* Studying new model architecture designs requires costly pretraining. Grafting materializes new architectures by editing pretrained models under small compute budgets (Sec. 3). *(c) Class-conditional image generation.* Samples generated by hybrid architectures obtained via grafting (Sec. 4). *(d) High-resolution text-to-image generation.* 2048×2048 samples generated using a grafted model (Sec. 5). *(e) Depth → width case study.* Samples generated using a model restructured via grafting (depth: 28 → 14) (Sec. 6).

A pretrained model implements a computational graph to perform tasks such as image or video generation. Given a new architectural idea and a pretrained model, we investigate whether the idea can be materialized by modifying its computational graph under small compute budgets. For example, one might hypothesize that a convolutional design could replace Multi-Head Attention (MHA) or Multi-Layer Perceptron (MLP) in a DiT. A simple way to materialize this idea is to replace MHA or MLP operators with a convolutional operator, while preserving model quality. This raises two key questions: (Q1) *operator initialization*: How to initialize a new operator before integrating it into the computational graph? (Q2) *error accumulation*: How to mitigate error propagation as multiple operators are integrated into the computational graph?

To address these questions, we present **grafting**[1], a simple two-stage approach to architecture editing (Fig. 1). Grafting proceeds as follows: (i) *activation distillation*: This stage transfers the functionality of the original operator to the new one by distilling its activations using a regression objective. (ii) *lightweight finetuning*: This stage mitigates error propagation caused by integrating multiple new operators by finetuning using limited data. Architectural editing spans multiple strategies—adding, removing, and replacing [5, 6, 7] operators. We focus on *operator replacement* as the core strategy: swapping one operator for another. Other strategies can be viewed as special cases of replacement.

The space of architectural editing is vast, raising a practical question: what types of replacements should we study? We first establish a *self-grafting baseline*, where we replace all MHA and MLP operators in DiT-XL/2 with randomly initialized counterparts. Despite the scale of this intervention, our grafting procedure achieves near-baseline model quality using under 1% of pretraining compute. Building on this, we focus on replacing existing operators with efficient alternatives, aiming to reduce model FLOPs while preserving quality. We also explore replacements that increase model FLOPs to examine broader design choices. To study this systematically, we construct a *testbed* based on DiT-XL/2 and define a set of architectural edits to evaluate how different grafting schemes affect model quality. We organize our design space along four axes: (1) which operator to replace (e.g., MHA, MLP); (2) what to replace it with (e.g., convolutions); (3) how to select layers for replacement (e.g., all layers); and (4) replacement ratio (full vs. partial). We focus on replacing MHA and MLP operators, as they account for a large fraction of model FLOPs. Replacements for MHA and MLP operators are motivated by empirical findings and prior architectural designs: our locality analysis supports local operators for MHA, while for MLP, we adopt ideas from prior work [8, 9, 10].

---

[1] Grafting draws inspiration from horticultural grafting, where efficient components (scions) are integrated into established systems (rootstock) to enhance functionality, such as yield and disease resistance [4].

We validate our grafting approach in increasingly challenging generative modeling setups:

**Result I: Grafting yields hybrid architecture designs with good quality for class-conditional image generation (Sec. 4.2).** We validate grafting using our testbed. For MHA (softmax attention), we explore several alternatives: local gated convolution (Hyena-SE, and our proposed Hyena-X/ Hyena-Y), local attention (sliding window), and linear attention (Mamba-2). For MLPs, alternatives include MLPs with variable expansion ratio (ratios=3, 6), and a convolutional variant (Hyena-X). Interestingly, several interleaved hybrid architecture designs achieve FID scores between 2.38 and 2.64 (DiT-XL/2 256x256 baseline: 2.27), showing that grafting can construct good quality hybrids (Tab. 4) [2] . Grafting is simple and lightweight: each experiment completes in under 24 hours on 8×H100 GPUs, using less than 2% of pretraining compute.

**Result II: We construct efficient hybrid architectures for high-resolution text-to-image (T2I) generation via grafting (Sec. 5).** We validate grafting in a challenging, real-world setting: 2048×2048 resolution T2I generation using PixArt-$\Sigma$ (DiT) [11]. This setting reflects key challenges: it operates on long sequences (16,384 tokens), involves a multimodal setup with text conditioning, and lacks training data. We target MHA operators for grafting, as they account for over 62% of generation latency. Using 12k synthetic data, our grafted model achieves a 1.43× speedup with <2% drop in GenEval score (47.78 vs. 49.75), showing that grafting scales to high-resolution, T2I generation.

**Case Study: Converting model depth to width via grafting (Sec. 6).** Motivated by our MLP grafting results, we rewire DiT-XL/2 by parallelizing every pair of transformer blocks, as modern GPUs favor parallel over sequential computation. This reduces model depth by 2× (28→14). The grafted model achieves FID=2.77, outperforming other models of comparable depth. To our knowledge, this is the *first attempt* to convert sequential transformer blocks into parallel in pretrained DiTs, enabling architectures to be restructured.

## 2    Prerequisites

**Diffusion models (DMs).** DMs generate data samples by iteratively denoising random noise. This sampling process inversely mirrors the forward data corruption mechanism: $\mathbf{z}_t = \alpha_t \mathbf{z} + \sigma_t \boldsymbol{\epsilon}$ where $\mathbf{z} = E(\mathbf{x}) \sim q(\mathbf{z})$ with $E$ representing a pretrained encoder and $\mathbf{x}$ the data variable. The noise term $\boldsymbol{\epsilon}$ follows the prior distribution $\mathcal{N}(0, I)$. The transition kernel from time 0 to $t$ is given by $q_t(\mathbf{z}_t|\mathbf{z}) = \mathcal{N}(\mathbf{z}_t; \alpha_t \mathbf{z}, \sigma_t^2 I)$. The choice of $\alpha_t$ and $\sigma_t$ defines the diffusion variant, such as variance-preserving [12], or flow matching [13]. The training objective [12] is as follows:

$$\mathcal{L}_{DM}(\boldsymbol{\phi}) = \mathbb{E}_{q(t)q(\mathbf{z},\mathbf{c})\mathcal{N}(\boldsymbol{\epsilon};0,I)}[\|\boldsymbol{\epsilon} - \boldsymbol{\epsilon}_{\boldsymbol{\phi}}(\mathbf{z}_t, t, \mathbf{c})\|_2^2], \tag{1}$$

where $q(t)$: time sampling distribution, and $q(\mathbf{z}, \mathbf{c})$: joint distribution of latent $\mathbf{z}$ and condition $\mathbf{c}$.

**Diffusion transformers (DiTs).** DiTs model the diffusion process by patchifying the input—noised images or latent—into a sequence of 1D tokens with positional embeddings. These tokens are processed through transformer blocks comprising self-attention, feedforward layers, residual connections, and normalization layers. DiTs also incorporate conditioning signals, such as noise timestep ($t$), class labels ($c$), or natural language prompts, enabling controllable generation [1, 14].

**Datasets and evaluation metrics.** For class-conditional image generation, we use ImageNet-1K [15]. We follow [1] and report Inception Score (IS), FID, sFID, Precision, and Recall using 50k generated samples (250 steps DDPM, cfg=1.5). For text-to-image generation, we report GenEval score [16].

## 3    Grafting Diffusion Transformers

### 3.1    Two-Stage Grafting Approach

Grafting aims to materialize new architectures by editing a pretrained model's computational graph. Given that we focus on replacing existing operators with alternatives, this raises two questions:

*(Q1) How should a new operator be initialized before being integrated into the computational graph?*
**Stage 1: Activation distillation.** We cast initialization as a regression task. Operators in a DiT block process $[B, N, D]$ inputs (batch, sequence, hidden) and output tensors of the same shape. Given a

---

[2] Strictly speaking, variable expansion ratio MLPs constitute a heterogeneous design rather than a hybrid (i.e. they do not introduce a new operator class); we use 'hybrid' throughout the paper for simplicity.

pretrained operator $f_\phi^l$ at layer $l$, we learn a new operator $g_\theta^l$ that approximates $f_\phi^l$ [17]. Since DiT activations are continuous and smooth, this can be posed as a regression problem:

$$\mathcal{L}(\boldsymbol{\theta}) = \mathbb{E}_{q(t)q(\mathbf{z},\mathbf{c})q_t(\mathbf{z}_t|\mathbf{z})}\big[\mathcal{L}_{\text{reg}}(g_{\boldsymbol{\theta}}^l(\mathbf{z}_t, t, \mathbf{c}), f_\phi^l(\mathbf{z}_t, t, \mathbf{c}))\big] \tag{2}$$

where $q(\mathbf{z}, \mathbf{c})$ is the joint distribution of latent representation $\mathbf{z}$ and condition $\mathbf{c}$, $q(t)$ is the time sampling distribution, and $q_t(\mathbf{z}_t|\mathbf{z})$ is the transition kernel from time 0 to $t$. $\mathcal{L}_{reg}$ is a regression objective such as $L_2$. In practice, a good initialization requires as few as 8k samples.

*(Q2) How can we mitigate error propagation as multiple operators are integrated into the computational graph?* **Stage 2: Lightweight finetuning.** As more operators are replaced, initialization errors propagate, leading to deviations from the pretrained model's behavior. We apply end-to-end finetuning with limited data to mitigate cumulative errors from stage 1. The fine-tuning objective is given in Equation 1. In practice, we find that competitive performance can be recovered using only 10% of the training data, even when replacing all MHA or MLP layers in DiT-XL/2.

## 3.2 Self-grafting Baseline

Prior to studying new architectural designs, we introduce *self-grafting*, a simple control setup where existing operators (e.g., MHA, MLP) are replaced with *identical* operators whose weights are randomly initialized. This preserves the computational graph's structure—operator types, receptive fields, and parameter count—while altering the computation performed. Self-grafting serves three purposes: (1) to assess the grafting procedure without architectural changes, (2) to provide a baseline for comparing replacements, and (3) to study factors affecting performance, such as data scale, regression objectives, and hyperparameters.

## 3.3 Activation Behavior Analysis and Self-grafting Results

We begin by analyzing the activation behavior of MHA and MLP operators across all layers in DiT-XL/2. In both cases, we observe large variance in activation values, particularly in deeper layers (Tab. 1 (i, ii)). When using regression-based distillation for Stage 1, these outliers affect optimization, particularly under the commonly used $L_2$ objective which penalizes all errors quadratically. This motivates a closer look at regression objectives. We study three regression objectives with different level of sensitivity to outliers—$L_2$, $L_1$, and Huber [18]—using a self-grafting setup. We select five representative layers ($l = 1, 8, 17, 27, 28$) for both MHA and MLP, spanning a range of activation

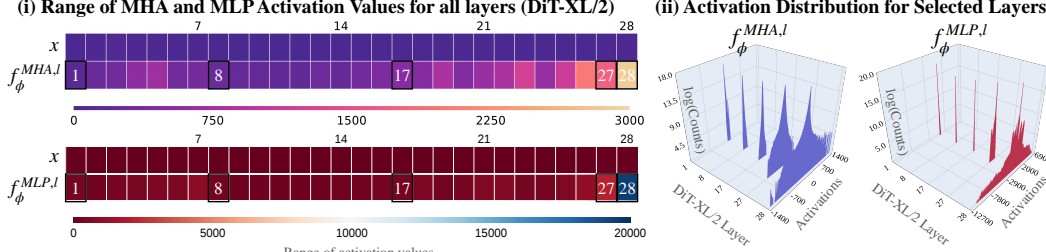

**(iii) MHA initialization**

| | IS ↑ | FID ↓ | sFID ↓ | Prec. ↑ | Rec. ↑ |
|---|---|---|---|---|---|
| Baseline | 278.20 | 2.27 | 4.60 | 0.83 | 0.57 |
| Random Init. | 40.86 | 76.27 | 10.33 | 0.33 | 0.52 |
| L2 | 269.31 | 2.58 | 5.75 | 0.82 | 0.58 |
| Huber ($\delta$=1.0) | 271.30 | 2.55 | **5.44** | 0.82 | 0.57 |
| **L1** | **273.03** | **2.51** | 5.48 | **0.83** | **0.58** |

**(iv) MLP initialization**

| | IS ↑ | FID ↓ | sFID ↓ | Prec. ↑ | Rec. ↑ |
|---|---|---|---|---|---|
| Baseline | 278.20 | 2.27 | 4.60 | 0.83 | 0.57 |
| Random Init. | 2.18 | 297.34 | 161.76 | 0.01 | 0.00 |
| **L2** | **265.34** | **2.33** | **4.38** | **0.81** | **0.59** |
| Huber ($\delta$=1.0) | 262.93 | 2.38 | 4.49 | 0.81 | 0.59 |
| L1 | 235.54 | 2.83 | 4.69 | 0.77 | 0.61 |

Table 1: **Activation statistics and self-grafting (Stage 1) results (DiT-XL/2). (i)** *Activation ranges (max–min)* across all 28 MHA and MLP operators, computed using 1,000 samples. Deeper layers exhibit higher variance in activation values. **(ii)** *Activation distributions (log-scale histograms)* for five selected layers (1, 8, 17, 27, 28), used in our initialization study. MLP layers show higher variance in activations than MHA, especially in deeper layers. **(iii, iv)** *Stage 1 results* for these layers using L2, Huber, and L1 regression. L1 yields the best FID for MHA (2.51), while L2 performs best for MLP (2.33), which contains 2× more parameters than MHA (10.6M vs. 5.3M). This study shows that high-quality initialization can be achieved by choosing operator-specific regression objectives.

values. Each operator is trained with 8K ImageNet-1K [15] samples, for 200 epochs with batch size 64 and learning rate 1e−4. We use $\delta = 1.0$ for Huber objective. We then integrate the initialized operators into the pretrained DiT-XL/2 and evaluate quality without any finetuning.

**High-quality initialization can be achieved by choosing operator-specific regression objectives.**

As shown in Tab. 1 (iii,iv), the choice of the regression objective affects performance. For MHA, $L_1$ achieves the best FID (2.51), followed by Huber (2.55) and $L_2$ (2.58). For MLPs, $L_2$ performs best (2.33), while $L_1$ underperforms (2.83); notably, MLPs have 2× more parameters than MHA which explains its robustness to outliers [19]. This shows that high-quality initialization requires tailored, activation-aware strategies. Further, we evaluate validation loss on held-out samples. For MHA, $L_1$ achieves the lowest loss; for MLPs, $L_2$ achieves the lowest loss for all blocks (See Sec. C.3).

| Stage 1 | Stage 2 | IS ↑ | FID ↓ | sFID ↓ | Prec. ↑ | Rec. ↑ |
|---|---|---|---|---|---|---|
| Baseline | | 278.20 | 2.27 | 4.60 | 0.83 | 0.57 |
| **MHA (Full Self-grafting)** | | | | | | |
| Random Init. | | 1.66 | 289.23 | 154.00 | 0.00 | 0.00 |
| 0.63% | – | 117.68 | 16.78 | 13.69 | 0.60 | 0.61 |
| 0.63% | 0.63% | 148.56 | 11.26 | 11.10 | 0.66 | 0.60 |
| 0.63% | 5.0% | 270.39 | 2.70 | 5.46 | 0.81 | 0.57 |
| **0.63%** | **10.0%** | **287.81** | **2.49** | **4.71** | **0.83** | **0.56** |
| **MLP (Full Self-grafting)** | | | | | | |
| Random Init. | | 1.27 | 314.72 | 204.99 | 0.00 | 0.00 |
| **0.63%** | **10.0%** | **277.72** | **2.54** | **4.52** | **0.83** | **0.57** |

Table 2: **Full self-grafting (Stage 2) results (DiT-XL/2).** We report results after replacing *all* 28 MHA and MLP operators using different amounts of training data. As we increase the training data from 0.63% (8k) to 10.0% (128k), FID improves consistently. Using only 10% of the training data, near-baseline performance is achieved: FID 2.49 for MHA and 2.54 for MLP.

**Full self-grafting with 10% data achieves near-baseline performance.** We extend our study to replace *all* MHA and MLP operators in DiT-XL/2 under the self-grafting setup and evaluate the effect of data on recovery (Tab. 2). For MHA, replacing all 28 layers without adaptation results in a noticeable performance drop, but Stage 2 (lightweight fine-tuning) is highly effective: using just 10% of the training data (128k samples), we achieve an FID of 2.53 vs. 2.27 for the baseline. Similarly, full MLP self-grafting with 10% data yields an FID of 2.54. We use batch size 256, learning rate $1e^{-4}$, and 30k iterations. In both cases, the quality is within 0.3 FID of the baseline, showing that full self-grafting is feasible under modest data and compute budgets.

### 3.4 Locality Analysis of Self-attention

MHA scales quadratically with sequence length, making it a computational bottleneck. A natural idea is to replace it with local operators, such as convolution or local attention. However, this will fail if the model relies on long-range dependencies: for example, replacing all MHA operators in DiT-XL/2 with a sliding window attention degrades FID from 2.27 to 53.9. To guide grafting, we quantify MHA locality using a simple band-$k$ metric. Given an attention matrix $A \in \mathbb{R}^{N \times N}$ and a band size of $k$, we define a bi-directional band indicator matrix $B_k \in \mathbb{R}^{N \times N}$ as:

$$(B_k)_{i,j} = \begin{cases} 1, & \text{if } |i-j| \leq k \\ 0, & \text{otherwise} \end{cases}$$

Then, locality within a band of size $k$ is computed as:

$$L_k = \frac{1}{N} \sum_{i,j} A_{i,j}(B_k)_{i,j} \quad (3)$$

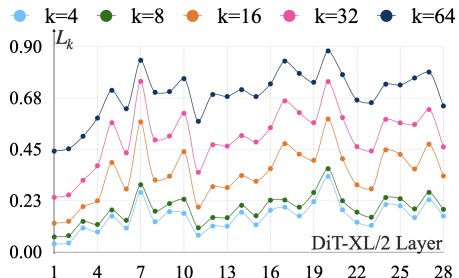

Figure 2: **Locality of self-attention in DiT-XL/2.** We plot $L_k$ values for all 28 MHA operators, averaged over timesteps and samples. At $k$=32, 15 out of 28 layers exhibit values exceeding 0.5, indicating that several MHA operators model local interactions.

We compute $L_k$ for all 28 MHA operators in DiT-XL/2 using 50-step DDIM sampling (250 ImageNet samples, sequence length 256, cfg scale 1.5), averaging across timesteps and samples. As shown in Fig. 2, MHA is largely local: on average, for $k$=32, 15 out of 28 layers attend to more than 50% attention mass within the band. The first few layers ($l$=1,2) display non-local attention patterns. Our analysis provides guidance for replacing MHA operators with efficient local operators.

# 4 Experiments I: Hybrid Architectures via Grafting

## 4.1 Testbed and Experiment setup

Building on our self-grafting results, we now ask: can we maintain model quality when existing operators are replaced with efficient alternatives? To investigate this, we study the grafting procedure along four design axes:

1. operator type to replace – MHA or MLP
2. replacement operator type – such as convolutions
3. layer selection strategy – replace operators in all layers or use heuristic-based selection
4. replacement ratio – full or partial

**We construct a testbed to systematically evaluate how design decisions affect generative quality under grafting.** We focus on efficient replacements that reduce FLOPs, but also include higher-FLOP variants to explore a broader range of architectural edits. We target MHA and MLP operators, which account for a significant portion of FLOPs in DiTs compared to other operators (e.g., normalization, activation, residuals). The rationale for replacing MHA or MLP operators is grounded in both empirical and architectural considerations: for MHA, our attention locality analysis (Fig. 2) motivates the use of local operators; for MLP, we leverage prior architecture ideas [8, 20, 21, 22, 9, 10]. Given a replacement operator, the decision to graft it to a model with $L$ transformer layers spans a space of $2^L$ configurations. To make this tractable, we study two layer selection strategies: full (replace all operators) and interleaved (replace operators in a repeating pattern) strategies. The latter is inspired by striped transformer designs [23, 24, 25]. Our testbed is detailed in Tab. 3.

**We introduce Hyena-X and Hyena-Y—two new efficient gated convolution operators designed as drop-in replacements for MHA.** While our study includes several off-the-shelf efficient alternatives, we also contribute new operator designs motivated by our MHA locality analysis. This allows us to test novel architectural ideas via grafting, broadening our study. Both Hyena-X and Hyena-Y are local gated convolutions composed of dense, short causal depth-wise 1D convolutions. Fig. 3 (left) illustrates their structure. We also adapt Hyena-X as an MLP alternative by applying it along the channel dimension. Hyena-X and Hyena-Y scale linearly with sequence length, compared to the quadratic scaling of MHA. Operator details are provided in Sec. G. We provide FLOP calculation for both operators in Sec. H.1.

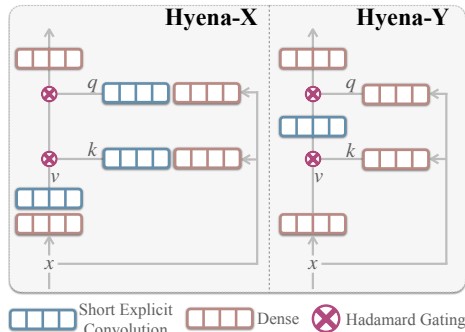

Figure 3: Our proposed Hyena-X and Hyena-Y, efficient local gated convolution operators used as drop-in replacements for MHA.

**Experiment setup.** For our hybrid experiments, we mostly use the hyperparameters determined from our self-grafting studies (Sec. 3.2).

*Stage 1: Operator initialization.* For each new operator, we perform activation distillation using 8K ImageNet-1K samples. Each operator is trained for 200 epochs with a batch size of 64 and an initial learning rate of $1e{-}4$. We pre-extract and store all regression features. All operators can be initialized in parallel. Each operator's training completes in under 30 minutes on a single H100 GPU. Experiment details for stage 1 are included in Sec. C.1.

*Stage 2: Lightweight finetuning.* For all experiments in Table 3, we use 10% of the ImageNet-1K training data and train for 50K steps. We use a batch size of 256, linearly warming up the learning rate to $1e{-}4$ over 1000 steps. Experiments typically complete in under 10 hours on 8×H100 GPUs. For specific ablations on increasing data, such as those involving 20% data or 100K steps, runtimes extend up to 24 hours (<2% pretraining compute). We provide experiment details in Sec. C.1.

| Operator Type: *Which operator types are we replacing?* | | |
|---|---|---|
| MHA, MLP | | |
| **Efficient Alternative:** *What do we replace it with?* | | |
| MHA | Convolutions: Hyena-SE [24], Hyena-X/ Hyena-Y *(Ours)* | $K$=4, causal |
| | Local Attention: Sliding Window Attention (SWA) [26, 27] | $w$=4, bidirectional |
| | Linear Attention: Mamba-2 [28] | $d_s$=64, $E$=2 |
| MLP | Variable expansion ratio | $r$=3,6 |
| | Hyena-X *(Ours)* | $r$=2, $K$=4, causal, mix channels |
| **Layer Selection:** *In which layers is the operator replaced?* | | |
| Full | Replace the operator in all layers | |
| Interleaved | Replace the operator in a repeating pattern (e.g., every 2 or 3 out of 4) | |
| **Replacement Ratio:** *What percentage of operators are replaced?* | | |
| 50%, 75%, 100% | | |

Table 3: **Grafting testbed with configurations.** This table defines the core design axes used in our study: operator type, efficient alternatives, layer selection strategy, and replacement ratio. For each alternative, we report configurations, including kernel size $K$, window size $w$, state size $d_s$, expand factor $E$, and MLP expansion ratio $r$. The baseline DiT-XL/2 operator uses $H$=16 attention heads and MLP expansion ratio $r$=4. Operator variants marked with (Ours) are proposed in this work.

## 4.2 Results and Insights

**MHA results.** Replacing MHA operators in DiT-XL/2 via grafting yields strong quality-efficiency tradeoffs. We discuss our key insights below:

- *Surprising effectiveness of operators with smaller receptive fields under interleaved grafting.* Our findings highlight that at 50% interleaved replacement, several alternatives—including SWA, Hyena-X/Y, and Mamba-2—consistently achieve FID scores within 0.5 of the baseline (2.27). The minimal FID drop observed especially with the SWA and Hyena variants, despite their limited receptive field ($K$=4, $w$=4), aligns with our locality analysis (Section 3.4).

- *Replacement strategy: Interleaved vs. Full.* Performance generally declines when increasing interleaved replacement from 50% to 75%. However, SWA remains effective at 75% interleaved replacement (FID=3.09). At 100% replacement, performance sharply degrades (all FIDs > 75). This trend aligns with our locality analysis, indicating that only a subset of layers are local and amenable to grafting.

- *Ablations on data scale and layer selection.* We study two factors under 50% MHA replacement. (i) Increasing fine-tuning data from 10% to 20% improves FID across all variants (e.g., Hyena-X: 2.74 → 2.61; SWA: 2.67 → 2.62, Mamba-2: 2.65→ 2.55) (Fig. 4 (a)). (ii) Under 50% replacement, we compare Hyena-X (interleaved) to three targeted heuristics: top-local (layers with highest band-$k$ values), low-local (layers with lowest band-$k$ values), and deep (last 14 layers). Interleaved yields the best FID (2.74), followed by top-local (3.02), low-local (3.18), and deep (4.00). These results confirm that interleaving is effective, and our band-$k$ metric identifies layers that are more amenable to grafting (Fig. 4 (b)).

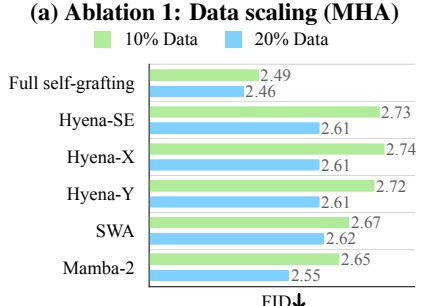

**(a) Ablation 1: Data scaling (MHA)**

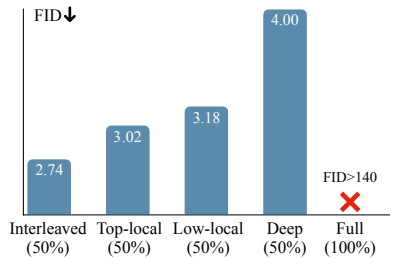

**(b) Ablation 2: Layer selection heuristics (MHA / Hyena-X)**

Figure 4: **Ablation studies.** *(a) Data scale:* Increasing fine-tuning data from 10% to 20% improves FID. *(b) Layer selection strategies:* Interleaved replacement outperforms other heuristics.

**MLP results.** Replacing MLP operators via grafting is effective. We discuss our key insights below:

- *Variable expansion ratio MLPs are effective under full replacement.* MLP alternatives with expansion ratio $r$=3 and $r$=6 demonstrate good quality under all replacement ratios. Even under full (100%) replacement, both variants maintain good performance, with $r$=3 achieving FID=2.66. This highlights that MLP width is a robust dimension for grafting.

- *Convolutional alternatives.* Hyena-X which combines dense and local channel mixing, performs competitively at 50% replacement (FID=2.63) but degrades at higher ratios, suggesting that such operators are only effective at moderate ratios.

> *Takeaway 1: Grafting is effective for constructing efficient hybrid architectures with good generative quality under small compute budgets. Interleaved designs are particularly effective.*

| | Ratio | IS ↑ | FID ↓ | sFID ↓ | Prec. ↑ | Rec. ↑ | $\Delta$**FLOPs**$_{op.}$ ↓ | $\Delta$**FLOPs**$_{ft.}$ ↓ | $\Delta$**Param** ↓ |
|---|---|---|---|---|---|---|---|---|---|
| Baseline | – | 278.20 | 2.27 | 4.60 | 0.83 | 0.57 | — | — | — |
| **MHA Grafting** | | | | | | | | | |
| Random Init. | 100% | 1.66 | 289.23 | 154.00 | 0.00 | 0.00 | — | — | — |
| Self-grafting | 100% | 287.81 | 2.49 | 4.71 | 0.83 | 0.56 | — | — | — |
| Hyena-SE ($K$=4) | 50% | 274.73 | 2.73 | 5.05 | 0.82 | 0.56 | -49.52% | +0.13% | +0.22% |
| | 75% | 231.15 | 3.62 | 6.04 | 0.81 | 0.54 | -74.27% | +0.20% | +0.33% |
| | 100% | ✗ | ✗ | ✗ | ✗ | ✗ | -99.03% | +0.26% | +0.43% |
| Hyena-X ($K$=4) | 50% | 273.30 | 2.74 | 5.03 | 0.83 | 0.56 | -49.90% | +0.13% | +0.16% |
| | 75% | 229.11 | 3.69 | 6.10 | 0.81 | 0.53 | -74.85% | +0.20% | +0.24% |
| | 100% | ✗ | ✗ | ✗ | ✗ | ✗ | -99.81% | +0.26% | +0.33% |
| Hyena-Y ($K$=4) | 50% | 273.37 | 2.72 | 5.02 | 0.83 | 0.55 | -49.52% | 0.00% | +0.05% |
| | 75% | 228.99 | 3.66 | 5.95 | 0.81 | 0.53 | -74.27% | 0.00% | +0.08% |
| | 100% | ✗ | ✗ | ✗ | ✗ | ✗ | -99.03% | 0.00% | +0.11% |
| SWA ($w$=4) | 50% | 280.62 | 2.67 | 4.90 | 0.83 | 0.56 | -48.24% | 0.00% | 0.00% |
| | 75% | 249.99 | 3.09 | 5.54 | 0.82 | 0.55 | -72.36% | 0.00% | 0.00% |
| | 100% | ✗ | ✗ | ✗ | ✗ | ✗ | -96.48% | 0.00% | 0.00% |
| Mamba-2 ($d_s$=64, $E$=2) | 50% | 285.08 | 2.65 | 4.84 | 0.83 | 0.55 | -37.59% | +77.89% | +28.02% |
| | 75% | 257.66 | 3.02 | 5.48 | 0.82 | 0.53 | -56.38% | +116.83% | +42.04% |
| | 100% | ✗ | ✗ | ✗ | ✗ | ✗ | -75.17% | +155.77% | +56.05% |
| **MLP Grafting** | | | | | | | | | |
| Random Init. | 100% | 1.27 | 314.72 | 204.99 | 0.00 | 0.00 | — | — | — |
| Self-grafting | 100% | 277.72 | 2.54 | 4.52 | 0.83 | 0.57 | — | — | — |
| Exp. ratio ↓ ($r$=3) | 50% | 272.14 | 2.53 | 4.51 | 0.83 | 0.57 | -12.50% | 0.00% | -12.50% |
| | 75% | 279.72 | 2.61 | 4.61 | 0.83 | 0.56 | -18.75% | 0.00% | -18.75% |
| | 100% | 252.11 | 2.66 | 4.57 | 0.81 | 0.57 | -25.00% | 0.00% | -25.00% |
| Exp. ratio ↑ ($r$=6) | 50% | 278.00 | 2.38 | 4.50 | 0.83 | 0.58 | +25.00% | 0.00% | +25.00% |
| | 75% | 277.94 | 2.37 | 4.48 | 0.82 | 0.58 | +37.50% | 0.00% | +37.50% |
| | 100% | 276.86 | 2.42 | 4.50 | 0.82 | 0.58 | +50.00% | 0.00% | +50.00% |
| Hyena-X ($r$=2,$K$=4) | 50% | 265.60 | 2.64 | 4.66 | 0.83 | 0.56 | +0.01% | 0.00% | +0.02% |
| | 75% | 226.13 | 3.26 | 4.79 | 0.81 | 0.55 | +0.02% | 0.00% | +0.03% |
| | 100% | ✗ | ✗ | ✗ | ✗ | ✗ | +0.02% | 0.00% | +0.03% |

Table 4: **Generation quality and efficiency metrics for MHA and MLP grafting.** We report quality (IS, FID, sFID, Precision, Recall) and efficiency ($\Delta$FLOPs and $\Delta$Param) results. Baseline refers to DiT-XL/2. For each alternative, setups that maintain FID within 0.5 of the baseline and offer the largest FLOPs reduction (or smallest FLOPs increase) are highlighted. ✗ denotes setups with poor generation (FID > 50). $\Delta$FLOPs and $\Delta$Param denote the percentage change in operator FLOPs and parameters, respectively. For MHA, total cost is split into $\Delta$FLOPs$_{op.}$ (softmax attention, gating) and $\Delta$FLOPs$_{ft.}$ (QKV/output projections, featurizers). We do not use this decomposition for MLP variants. Mamba-2 incurs higher $\Delta$FLOPs$_{ft.}$ due to additional projections. FLOP expressions are provided in Sec. H.1. **Key result:** Many interleaved designs achieve good quality generation (FID within 0.5 of baseline). All experiments use 10% training data and <1% pretraining compute.

# 5 Experiments II: Grafting Text-to-Image Diffusion Transformers

We apply grafting to a more challenging setting: high-resolution text-to-image generation with PixArt-Σ [11]. This presents three challenges: (1) long sequences (16,384 tokens for 2048×2048 resolution), (2) a multimodal setup with text conditioning, and (3) lack of publicly available training data. These factors make PixArt-Σ a representative setting for evaluating grafting under real-world constraints. PixArt-Σ contains 28 transformer layers similar to DiT-XL/2.

**Experiment setup.** We replace MHA operators in PixArt-Σ with Hyena-X via grafting, as MHA accounts for over 62% of generation latency. Hyena-X was chosen based on its good quality-efficiency tradeoff in the ImageNet-1K setup, achieving FID 2.61 with 20% data (see Fig. 4 (b)). Interleaved grafting is applied for layers 8, 10, 12, 14, 16, 18, and 20–27; empirically, we found that layers 20–27 can be replaced without significant quality drop. For grafting, we created a small, uncurated synthetic dataset of 12k image-text pairs. The text prompts for this dataset were sampled from the 30k publicly released evaluation set. *Stage 1 (activation distillation):* 8k uncurated synthetic image-text pairs are used to initialize Hyena-X blocks. We use the L1 regression objective, as we observe similar MHA activation behavior in PixArt-Σ (Fig. E.1). *Stage 2 (finetuning):* We use LoRA (rank=64) [29] for finetuning. LoRA enables efficient finetuning by managing the high memory demands associated with long sequences (16,384 tokens). The full 12k synthetic dataset is used in this stage. We use 20 step DPM Solver [30] for generation. Experiment details are provided in Sec. E.2.

**Results.** The grafted model achieves a 1.43× speedup in wall-clock time, with a small drop in GenEval score (47.78 vs. 49.75). Attribute-specific metrics remain comparable, and qualitative samples show good alignment and quality. Some localized artifacts are observed in textured regions likely due to LoRA's adaptation capacity and low-quality synthetic data (see failure cases in Fig. E.3, E.4).

> *Takeaway 2: We graft high-resolution text-to-image DiTs, constructing hybrid architectures with meaningful speedups and minimal quality drop.*

| Model | Ratio | Obj(1) | Obj(2) | Count | Colors | Pos | Color Attr. | Overall ↑ | Latency (ms) ↓ |
|---|---|---|---|---|---|---|---|---|---|
| Baseline | - | 81.45 | 61.62 | 46.25 | 77.13 | 10.75 | 21.50 | 49.75 | 235.46 |
| Hyena-X | 29% | 80.31 | 59.34 | 49.69 | 68.62 | 11.50 | 18.75 | 48.04 | 194.95 (1.21×) |
| Hyena-X | 50% | 80.00 | 57.07 | 48.13 | 70.74 | 11.25 | 19.50 | **47.78** | **164.58** (1.43×) |

Table 5: **GenEval results and latency for PixArt-Σ and the grafted variants.** The 50% grafted model achieves a 1.43× speedup while retaining strong text-image alignment (GenEval overall score: 47.78 vs. 49.75). Attribute-specific scores remain comparable across models. Latency is measured for a single forward pass on an Nvidia H100 (batch size=2).

# 6 Case Study: Converting Model Depth to Width via Grafting

**Can we rewire two sequential transformer blocks to run in parallel?** Our MLP grafting results showed that MLPs are amenable to grafting, even at 100% replacement with an expansion ratio of $r = 6$, demonstrating that wider computation within an operator is feasible. This success, combined with the fact that modern GPUs favor parallel over sequential computation, motivates a broader question: can we convert deeper, sequential DiT computations into wider, parallel ones via grafting while maintaining quality? To explore this, we rewire DiT-XL/2 by parallelizing every pair of sequential transformer blocks—each pair receives the same input, and their outputs are merged via a linear projection. This reduces model depth by 2× (28 → 14) with a 6% increase in parameters.

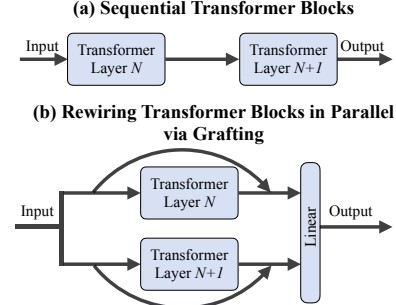

Figure 5: Convert model depth → width via grafting: (a) Two sequential transformer layers. (b) Rewiring in parallel via grafting (includes skip connections).

**Experiment Setup.** The rewiring schematic is shown in Fig. 5. We use DiT-XL/2. *Stage 1: Activation distillation.* Each parallel pair was initialized via activation distillation using L1 regression. The weights for each block in the parallel pair were initialized from their corresponding pre-trained weights, rather than random initialization. Similar to our previous experiments, 8k ImageNet-1K samples were used for this stage. *Stage 2: Lightweight finetuning.* Given the architectural restructuring, finetuning was performed using 25% of the training data. The learning rate was linearly warmed up to 1e-4 and halved at 75k and 150k iterations. Additional details can be found in Sec. D.

**Results.** The goal of this study is to evaluate generative quality (FID) vs. model depth. We report results in Tab. 6. To contextualize our findings, we compare against two categories: (i) DiTs trained from scratch at lower depth, and (ii) pruning methods [31, 32]. Our 14-layer grafted model achieves an FID of 2.77—surpassing DiT variants trained from scratch with similar or increased depth, including DiT-L/2 (depth 24, FID 3.73) and U-ViT-L (depth 21, FID 3.44). It also outperforms pruning baselines such as TinyDiT-D14 with masked knowledge distillation (depth 14, FID 2.86) and BK-SDM (depth 14, FID 7.43), though these baselines have fewer parameters (340M) compared to the grafted variants (712M).

> *Takeaway 3: Grafting enables architectural restructuring at the transformer block level, allowing model depth to be traded for width.*

| Method | Depth | A.R | Iters | IS ↑ | FID ↓ | sFID ↓ | Prec. ↑ | Recall ↑ | Speedup ↑ | Params ↓ |
|---|---|---|---|---|---|---|---|---|---|---|
| DiT-L/2 [1] | 24 | 42.7 | 1,000K | 196.26 | 3.73 | 4.62 | 0.82 | 0.54 | — | 458M |
| U-ViT-L [33] | 21 | 48.8 | 300K | 221.29 | 3.44 | 6.58 | 0.83 | 0.52 | — | 287M |
| DiT-B/2 [1] | 12 | 64.0 | 1000K | 119.63 | 10.12 | 5.39 | 0.73 | 0.55 | — | 130M |
| BK-SDM [31] | 14 | 82.3 | 100K | 141.18 | 7.43 | 6.09 | 0.75 | 0.55 | 2× | 340M |
| TinyDiT-D14 [32] | 14 | 82.3 | 500K | 198.85 | 3.92 | 5.69 | 0.78 | 0.58 | 2× | 340M |
| TinyDiT-D14 w/ MKD [32] | 14 | 82.3 | 500K | 234.50 | 2.86 | 4.75 | 0.82 | 0.55 | 2× | 340M |
| DiT-XL/2 [1] | 28 | 41.4 | 7,000K | 278.20 | 2.27 | 4.60 | 0.83 | 0.57 | 1× | 675M |
| **Grafting (Ours)** | 14 | 164.6 | 100K | 231.91 | 3.12 | 4.71 | 0.82 | 0.55 | 2×¶ | 712M |
| **Grafting (Ours)** | 14 | 164.6 | 230K | 251.77 | 2.77 | 4.87 | 0.82 | 0.56 | 2×¶ | 712M |

Table 6: **Generative quality vs. model depth.** We report generative quality metrics (IS, FID, sFID, Precision, and Recall). A.R. (Aspect Ratio) is defined as model width divided by depth (e.g., 1152/14 = 82.3). For pruning and grafting setups, we report speedup with respect to DiT-XL/2 (depth=28). Off-the-shelf DiT-L/2, U-ViT-L, and DiT-B/2 scores, along with pruning baselines (BK-SDM, TinyDiT-D14, and TinyDiT-D14 w/ MKD), are sourced from [32]. MKD refers to Masked Knowledge Distillation, a recovery method used in [32]. ¶ Speedup is measured for a single forward pass on an Nvidia H100 (batch size=2). (details in Sec. D). **Key result.** Our grafted models achieve better generative quality at depth=14, surpassing baselines in FID, IS, Precision, and Recall.

# 7 Conclusion and Discussion

In this work, we introduced *grafting*, a simple approach to architecture editing. We constructed hybrid models by replacing self-attention and MLPs with efficient alternatives, achieving competitive quality (FID 2.38–2.64 vs. 2.27 baseline). We then applied grafting to a high-resolution text-to-image model (PixArt-Σ), yielding a 43% speedup with less than a 2% drop in GenEval score. We then used grafting to restructure DiT-XL/2, converting every pair of sequential transformer blocks into parallel, reducing model depth by half and yielding better quality (FID 2.77) among 14-layer DiTs. These results demonstrate grafting's utility in both short- and long-context settings (e.g., ImageNet-1K and PixArt-Σ, respectively), and for architecture restructuring. Overall, grafting proves to be a lightweight approach for materializing diffusion transformer designs under small compute budgets.

**Related work and discussion.** Due to page limit, we discuss related work in Supp. A. Further, to demonstrate the generalization of grafting, we graft an LLM (Qwen3-4B [34])—a generative model (autoregressive), model architecture, and data modality distinct from diffusion-based image generation with DiTs (Supp. F). We discuss broader impact, limitations and applications in Supp. I.

**Acknowledgments.** We thank Liquid AI for sponsoring compute for this project. We also thank Armin W. Thomas, Garyk Brixi, Kyle Sargent, Karthik Dharmarajan, Stephen Tian, Cristobal Eyzaguirre, and Aryaman Arora for their feedback on the manuscript.

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

# Supplementary Material

# A  Related Work

**Diffusion model architectures.** Recently, many architectural innovations have been proposed for diffusion models for image and video generation [35, 36, 37, 38, 39, 40, 41, 42, 43, 44]. Many recent works focus on improving the attention mechanism in diffusion models to enhance efficiency and scalability. One major direction is the use of modern linear attention variants, such as DiffuSSM [36], DiS [37], Zigma [38], DiM [39], and DIG [40]. Recently, text-to-image diffusion models such as SANA [35] have also adapted linear attention variants to support high-resolution generation. Another recent direction explores the mixture-of-experts (MoE) idea. DiT-MoE [45] introduces sparse diffusion transformers with shared expert routing and expert-level balance loss, enabling efficient scaling to 16.5B parameters while achieving competitive performance. We note that methods like STAR [46] have also successfully discovered architectures via evolutionary methods for autoregressive language modeling. Recently, Bian *et al.* [47] showed that pretraining wider and shallower autoregressive language models can yield efficiency gains while preserving accuracy. While effective, these approaches require training from scratch, making such studies expensive and inaccessible to practitioners. In contrast, grafting focuses on architecture editing of pretrained models to materialize new architectures under small compute budgets.

**Architectural editing of pretrained generative models.** Another line of work focuses on linearizing large language models by replacing softmax attention with efficient operators, such as linear attention [5, 6, 7]. Similar ideas have also been adopted for diffusion models in [48, 49, 50], though these works focus only on ultra-high-resolution settings. These prior efforts typically focus on replacing a single operator type (primarily attention) or are limited to specific application domains. Grafting presents a more general and comprehensive approach for architectural editing. It extends beyond single-operator replacement to enable modifying multiple operator types, exploring diverse architectural alternatives (e.g., both MHA and MLP replacements), and restructuring architectures (e.g., converting model depth to width). Recently, FFN Fusion [51] explored parallelizing transformer blocks in LLMs, aiming to reduce sequential computation. While our work focuses on diffusion-based generative modeling with transformers, prior work on image classification has explored model reuse and modification, such as few-shot knowledge distillation via progressive network grafting [52] and modular recombination of pretrained model components [53].

# B    Standard Deviation of Experiments

To compute variance associated with our reported results, we repeat two representative experiments—MHA (Hyena-Y) and MLP (width=6)—using three different random seeds (`seed = 0, 200, 300`). We follow the exact grafting setup used in the main paper for these experiments. We report the mean and standard deviation of IS, FID, sFID, Precision and Recall in Tab. B.1. We observe that the standard deviations are within an acceptable range.

| Setup | IS | FID | sFID | Precision | Recall |
|---|---|---|---|---|---|
| **MHA/ Hyena-Y** | $273.19 \pm 0.46$ | $2.73 \pm 0.01$ | $5.06 \pm 0.04$ | $0.83 \pm 0.00$ | $0.55 \pm 0.00$ |
| **MLP/ higher width ($r = 6$)** | $277.91 \pm 0.95$ | $2.41 \pm 0.01$ | $4.48 \pm 0.02$ | $0.82 \pm 0.00$ | $0.58 \pm 0.00$ |

Table B.1: Mean and standard deviation of IS, FID, sFID, Precision and Recall calculated for three runs with different random seeds (`0,200,300`).

# C    Hybrid Architecture Experiments: Additional details

## C.1    Experiment details and additional samples

We provide all hyperparameters used for the experiments in Tab. C.1. To ensure a fair comparison, we used identical hyperparameters across every hybrid experiment. We include additional qualitative samples generated using our hybrid architectures obtained via grafting in Fig. C.1.

| **Stage 1: Activation Distillation** | |
|---|---|
| Initial Learning Rate | $1 \times 10^{-3}$ |
| Weight Decay | 0 |
| Epochs | 200 |
| Batch Size | 64 |
| Clip Norm | 10.0 |
| Optimizer | AdamW ($betas = (0.9, 0.999)$) |
| Loss Function | L1 (MHA), L2 (MLP) |
| **Stage 2: Lightweight Finetuning** | |
| Initial Learning Rate | $1 \times 10^{-4}$ |
| Weight Decay | $5 \times 10^{-5}$ |
| Iterations | 50,000 (100 epochs) |
| Batch Size | 256 |
| Optimizer | AdamW ($betas = (0.9, 0.999)$) |
| Scheduler | Linear Warmup over 1K steps, then constant lr |
| Training Data | 10% of ImageNet-1K (128k samples) |

Table C.1: Experiment details for MHA/MLP grafting experiments using DiT-XL/2 (ImageNet-1K).

## C.2    Modulated Regression Targets for MHA

For Stage 1, we explored a modulation-aware regression variant for MHA experiments that incorporates the learned scalar (`gate_msa`) applied to the attention output. In the standard setup, we regress from input $x$ to the raw output of the attention block $y = \texttt{MHA}(\cdot)$. In the modulation-aware formulation, the target becomes $y = \texttt{gate\_msa} \odot \texttt{MHA}(\cdot)$. Tab. C.2 compares these two variants with L1 and L2 loss. Modulation-aware regression increases target scale, which adversely affects L2 loss performance due to its sensitivity to large values. L1 performs similarly in both settings. We adopted the standard (modulation-agnostic) formulation for all experiments for simplicity.

| | Modulation-aware | IS | FID | sFID | Precision | Recall |
|---|---|---|---|---|---|---|
| Baseline | | 278.20 | 2.27 | 4.60 | 0.83 | 0.57 |
| L2 | ✓ | 246.17 | 3.00 | 7.11 | 0.79 | 0.58 |
| | ✗ | 269.31 | 2.58 | 5.75 | 0.82 | 0.58 |
| L1 | ✓ | 272.86 | 2.51 | **5.29** | 0.82 | 0.57 |
| | ✗ | **273.03** | **2.51** | 5.48 | **0.83** | **0.58** |

Table C.2: Comparison of modulation-aware and standard regression targets for Stage 1. The modulation-aware setup includes the learned scalar (`gate_msa`) as a multiplicative factor in the regression target. L2 loss is sensitive to the amplified target scale and performs worse, while L1 loss remains robust and performs similarly in both cases. We adopt the standard formulation by default.

## C.3 Validation Loss Curves for Self-grafting Experiments

To support the trends reported in the main paper (Sec.3.2, Table 2), we include validation loss curves in Fig.C.2 for five representative layers. Loss is computed using $L_2$ on a held-out set of 8k samples. For MHA layers (top row), $L_1$ consistently achieves lower validation loss in deeper layers, reflecting robustness to high activation variance. For MLP layers (bottom row), $L_2$ generalizes best across all layers. This contrast may be explained by parameter count: MLPs have roughly 2× more parameters than MHA layers, making them less sensitive to outliers and better suited to $L_2$ regression.

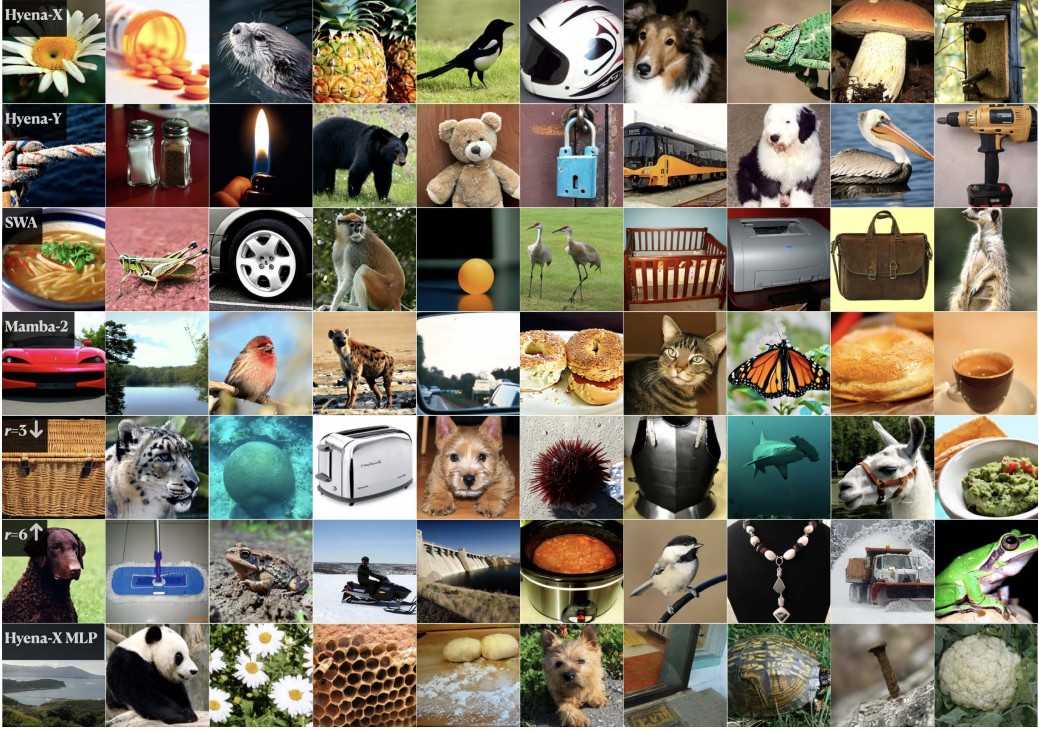

Figure C.1: **Additional samples generated by grafted DiT-XL/2 variants.** Each row corresponds to a different hybrid. We report FID scores (*lower is better*, ImageNet-1K 256×256) for each hybrid. *MHA variants (top 4 rows)*: Hyena-X (2.61), Hyena-Y (2.61), SWA (2.62), Mamba-2 (2.55). *MLP variants (bottom 3 rows)*: Lower width (2.53), Higher width (2.38), Hyena-X MLP (2.64). These results highlight the flexibility of grafting in constructing high-quality hybrid architectures by replacing MHA or MLP operators.

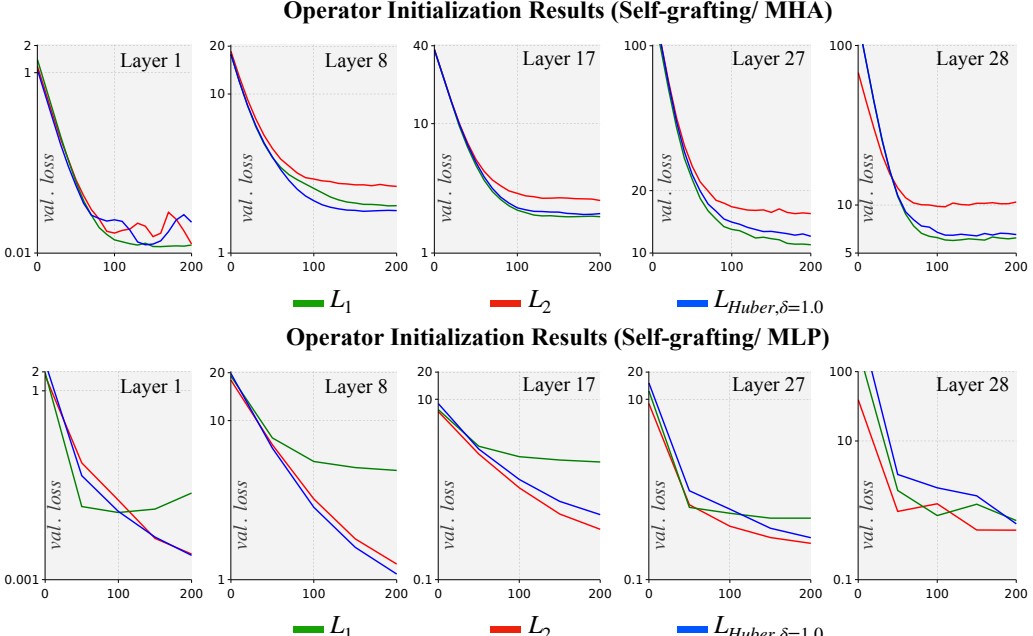

Figure C.2: Validation loss curves for MHA (top) and MLP (bottom) operator distillation showing the training dynamics for three regression objectives. As one can observe, L1 shows better generalization for MHA and L2 shows better generalization for MLP.

# D  Depth to Width Grafting Experiments: Additional details

We provide all hyperparameters used for these experiments in Tab. D.1. We show additional samples in Fig. D.1.

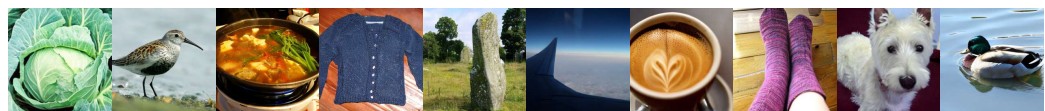

Figure D.1: **Depth-to-width grafting samples**. Samples from a DiT-XL/2 model in which every pair of transformer block is converted into a parallel block, effectively reducing depth by $2\times$ (FID=2.77).

**Implementation details.** The generative quality metrics (FID, IS, sFID, Precision, Recall) reported in Tab. 6 correspond to the exact implementation of rewiring presented in Fig. 5. For speedup measurements, a simple fused version of our rewired version was used. It is important to note that the reported speedup values are expected to decrease with large batch sizes, primarily due to the model parameter count (712M). Future work will focus on exploring hardware-aware / optimized implementations to achieve consistent speedup across a wider range of batch sizes.

# E  Text-to-Image Generation Experiments: Additional details

## E.1  MHA activation plots

We show activation distribution for five representative layers ($l = 15, 17, 19, 21, 23$) in Fig. E.1.

## E.2  Experiment details

We provide all hyperparameters used in our PixArt-$\Sigma$ grafting experiments in Tab. E.1.

| Stage 1: Activation Distillation | |
|---|---|
| Initial Learning Rate | $1 \times 10^{-4}$ |
| Regression Objective | L1 |
| Epochs | 200 |
| Batch Size | 64 |
| Optimizer | AdamW $(betas = (0.9, 0.999))$ |
| **Stage 2: Lightweight Finetuning** | |
| Learning Rate | $1 \times 10^{-4}$ |
| Weight Decay | 0 |
| Iterations | 230k |
| Batch Size | 256 |
| Optimizer | AdamW $(betas = (0.9, 0.95))$ |
| Scheduler | Warmup over 1K steps, half every 75k steps |
| Training Data | 25% of ImageNet-1K (320k) |

Table D.1: Experiment details for depth-to-width grafting experiments using DiT-XL/2 (ImageNet-1K).

**Activation Distribution for Selected Layers (PixArt $-\Sigma$)**

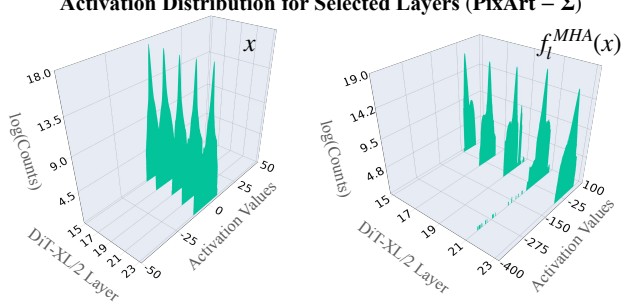

Figure E.1: Similar to DiT-XL/2, we observe high activation variance in PixArt-$\Sigma$ MHA operators. We show input (left) and output (right) activation values corresponding to five representative layers (15, 17, 19, 21, 23) in PixArt-$\Sigma$.

### E.3 Generated samples and failure cases

We show additional high-resolution samples generated by the grafted PixArt-$\Sigma$ model in Fig. E.2, illustrating the model's ability to preserve generative quality across diverse prompts despite substantial architectural edits. Figure E.3 illustrates two types of failure modes observed in grafted PixArt-$\Sigma$ outputs. Each column pair shows the output of PixArt-$\Sigma$ (left) and the grafted model (right) for the same text prompt. In the top row, the original model generates images that are reasonably aligned with the prompts, while the grafted model fails to preserve this alignment—indicating limitations during the LoRA-based finetuning stage. In the bottom row, the synthetic supervision itself is of low quality, resulting in poor outputs from both the original and grafted models. To better understand this issue, Figure E.4 presents additional examples of low-quality synthetic data produced by PixArt-$\Sigma$ and used for grafting. These samples often exhibit artifacts and unrealistic physics. While synthetic data enables low-cost adaptation, these results highlight the importance of improved data curation and filtering to avoid propagating errors during the grafting process.

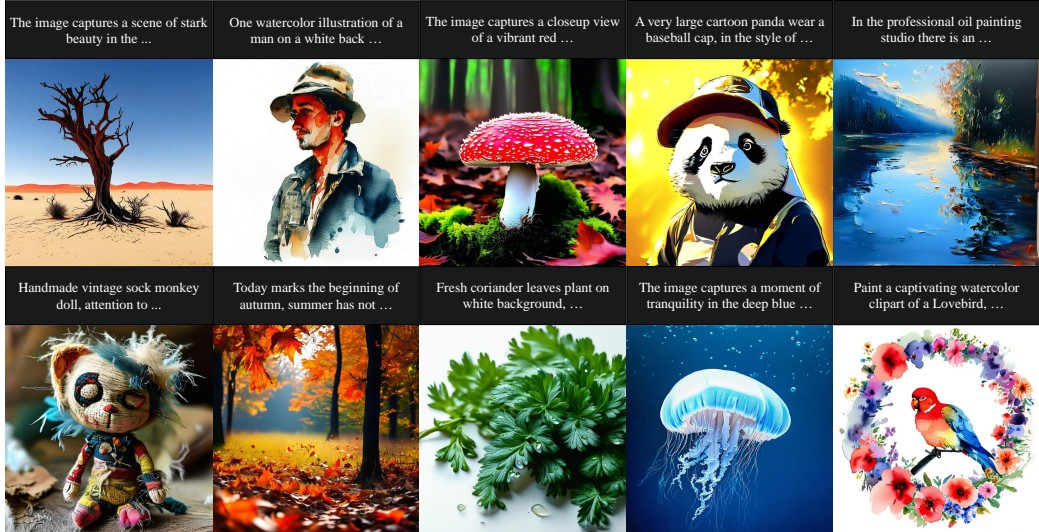

Figure E.2: Additional 2048×2048 samples generated using our grafted PixArt-Σ model.

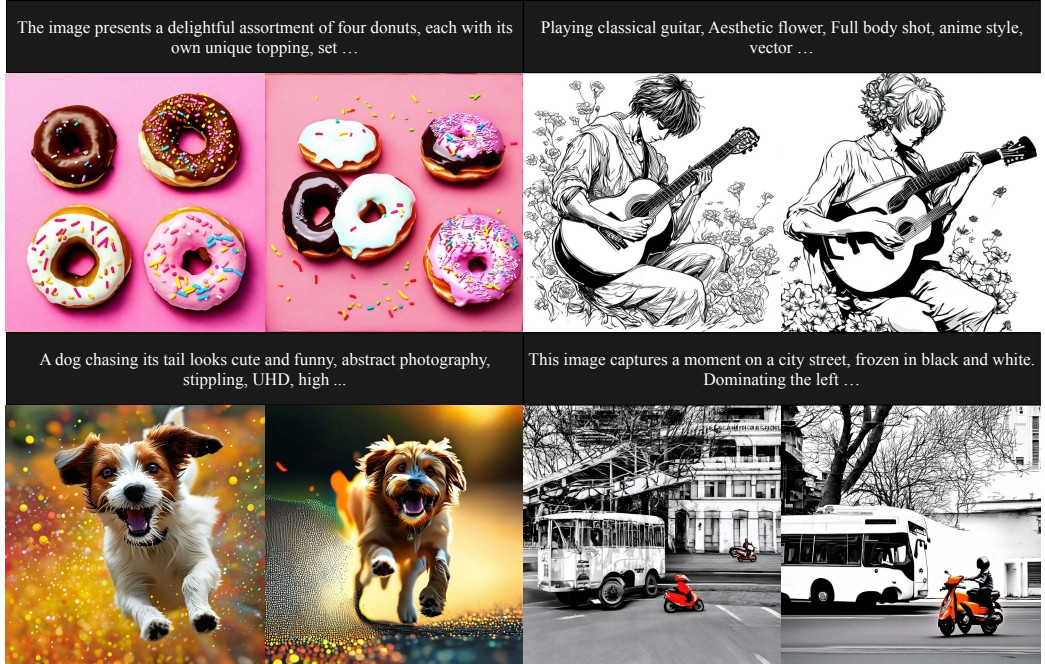

Figure E.3: **Text-to-image generation failure cases.** Each pair shows outputs from PixArt-Σ (left) and the grafted model (right) for the same prompt. In the top row, the prompt specifies four donuts with unique toppings and a full-body anime-style character playing classical guitar. The grafted outputs deviate from these prompts—showing incorrect object counts (e.g., five donuts) and degraded structure (e.g., distorted hands), reflecting text-image misalignment and visual artifacts introduced during grafting. In the bottom row, the supervision itself is poor: prompts such as a 'dog chasing its tail in UHD stippling style' and 'a black-and-white street photo' are not faithfully captured by either model. These examples highlight challenges arising both from LORA finetuning and low-quality synthetic data.

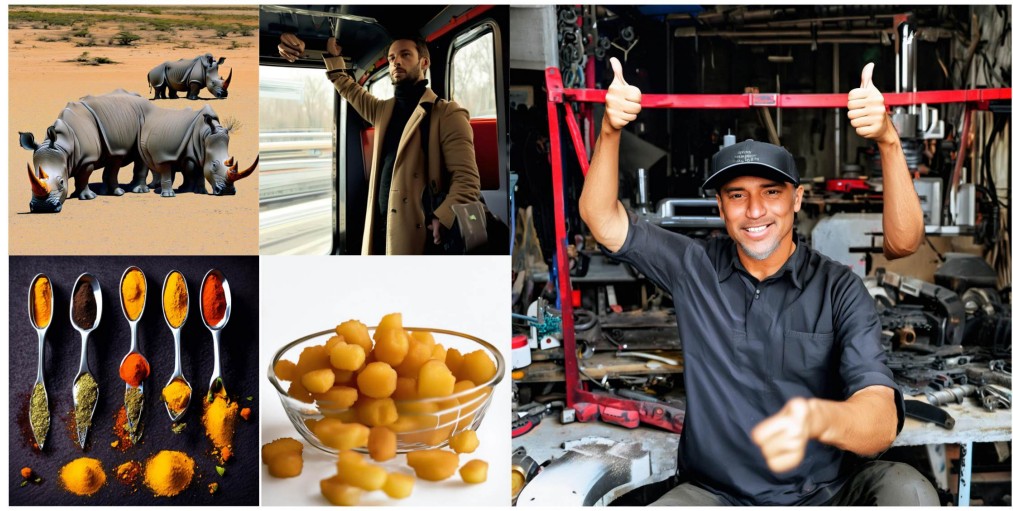

Figure E.4: **Examples of low-quality samples generated by PixArt-$\Sigma$ used for grafting**. These images contain unrealistic features, inconsistent physics, and visual artifacts. Their presence in the grafting dataset can degrade generation quality of grafted models, highlighting the importance of data curation when using synthetic data.

| Stage 1: Activation Distillation | |
|---|---|
| Initial Learning Rate | $1 \times 10^{-4}$ |
| Weight Decay | $1 \times 10^{-5}$ |
| Epochs | 100 |
| Clip Norm Value | 0.1 (Layers 20-27), 0.01 (Other layers) |
| Batch Size | 16 |
| Optimizer | AdamW |
| Scheduler | Half lr at epochs = 50 |
| **Stage 2: Lightweight Finetuning** | |
| Initial Learning Rate | $1 \times 10^{-5}$ |
| Weight Decay | 0 |
| Iterations | 18k |
| Batch Size | 64 (with gradient accumulation) |
| Optimizer | AdamW |
| Scheduler | linear warmup (500 steps), then constant lr |
| LoRA rank | 64 |

Table E.1: Experiment details for PixArt-$\Sigma$ grafting experiments.

## F    Grafting Autoregressive Large Language Models

To further validate the generality of grafting, we apply our two-stage procedure to a large language model (Qwen3-4B [34]) for autoregressive language modeling—a task, architecture, and modality distinct from diffusion-based image generation with DiTs.

**Setup.** We apply our two-stage grafting procedure to an autoregressive transformer, Qwen3-4B, for next-token prediction. This experiment replaces the Multi-Head Attention (MHA) operators with Sliding Window Attention (SWA) using a window size of 256. Stage 1 (activation distillation) and Stage 2 (lightweight fine-tuning) are performed identically to the diffusion experiments, with context lengths of 1024 and 8192 tokens respectively. We use 50k instruction-following examples from the Alpaca-cleaned dataset [3] following [5]. We evaluate on standard reasoning and commonsense benchmarks—PiQA [54], ARC-e [55], ARC-c [55], HellaSwag [56], Winogrande [57], and MMLU (5-shot) [58]—following the evaluation setup in [5].

**Results.** As shown in Table F.1, our grafted Qwen3-4B model achieves a $1.4\times$ decode throughput (8k context length) while maintaining average performance within 1% of the baseline. This result demonstrates that grafting extends beyond diffusion-based image generation, generalizing to new generative modeling task, model architecture, and data modality.

| Model | Stage | Ratio | PiQA | ARC-e | ARC-c | HellaSwag | Winogrande | MMLU | Avg. | Speedup ↑ |
|---|---|---|---|---|---|---|---|---|---|---|
| Baseline | — | — | 74.9 | 80.5 | 54.0 | 68.5 | 66.0 | 70.1 | 69.0 | — |
| Grafting (Ours) | Random Init. | 50% | 64.3 | 56.5 | 29.7 | 37.9 | 50.9 | 25.9 | 44.2 | 1.4× |
| Grafting (Ours) | Stage 1 | 50% | 74.8 | 80.4 | 53.4 | 68.2 | 65.5 | 66.9 | 68.2 | 1.4× |
| Grafting (Ours) | Stage 2 | 50% | 75.5 | 80.3 | 52.3 | 69.6 | 67.0 | 66.9 | 68.6 | 1.4× |

Table F.1: **Qwen3-4B [34] grafting results.** We evaluate grafting on Qwen3-4B by replacing Multi-Head Attention (MHA) with Sliding Window Attention (SWA, window=256). Stage 1 denotes activation distillation; Stage 2 denotes lightweight fine-tuning. Performance is reported on standard commonsense and reasoning benchmarks. Our grafted model achieves a $1.4\times$ decode throughput at 8k context length (single Nvidia H100) with less than a 1% drop in average performance relative to the baseline.

## G    Hyena-X and Hyena-Y operators: Additional details

Informed by our band-k analysis of MHA operators, we introduce a collection of efficient operators designed to exploit the locality in attention matrices. Given an input $x \in \mathbb{R}^{\ell \times d}$, a generic Hyena operator performs the following transformation:

$$q_s^c = \sum_{s'} T_{ss'}^c \sum_{c'} x_{s'}^{c'} W^{c'c}$$

$$k_s^c = \sum_{s'} H_{ss'}^c \sum_{c'} x_{s'}^{c'} U^{c'c}$$

$$v_s^c = \sum_{s'} K_{ss'}^c \sum_{c'} x_{s'}^{c'} P^{c'c}$$

$$y_s^c = \sum_{c'} \sum_{s'} (q_s^{c'} G_{ss'}^{c'} k_{s'}^{c'} v_{s'}^{c'}) M^{c'c}$$

where $W, U, P, M \in \mathbb{R}^{d \times d}$ are parametrized as dense or low-rank matrices, and $T, H, K, G \in \mathbb{R}^{\ell \times \ell}$ are Toeplitz matrices corresponding to convolutions with the filters $h_T, h_H, h_K, h_G$, respectively. In the original formulation [59], the filters $h_T, h_H, h_K$ are short and explicitly parametrized, whereas $h_G$ is implicitly parametrized.

We build on this formulation and propose `Hyena-X` and `Hyena-Y`, two Hyena operators designed for grafting. `Hyena-X` removes the implicit convolution entirely by setting $G = I$. In contrast, `Hyena-Y` introduces two changes: (i) it removes all three featurizer convolutions ($T, H, K$), and (ii) replaces

---

[3] https://huggingface.co/datasets/yahma/alpaca-cleaned

the implicit long convolution in $G$ with a short, explicit convolution. This modified structure preserves local inductive bias while significantly reducing computational cost. An illustration is provided in main paper. These operators allows us to realize speedups across a range of inputs resolutions: both `Hyena-X` and `Hyena-Y` are faster than Mamba-2 operators on all input sequence lengths, including lower resolution regimes.

# H  FLOP calculation

Notations are provided in Tab. H.1.

## H.1  MHA

- **Input projections (Q, K, V)**: $6LD^2$
- **Softmax attention**: $4L^2D + 2HL^2$
- **Output projection**: $2LD^2$

## H.2  SWA

- **Input projections (Q, K, V)**: $6LD^2$
- **Sliding window attention (Bidirectional)**: $4L(2w+1)D + 2HL(2w+1)$
- **Output projection**: $2LD^2$

## H.3  Hyena-SE

- **Input projections**: $6LD^2$
- **Featurizer**: $3LDK \times 2$
- **Inner filter convolution**: $LDK \times 2$
- **gates**: $LD \times 2$
- **Output projection**: $2LD^2$

## H.4  Hyena-X

- **Input projections**: $6LD^2$
- **Featurizer**: $3LDK \times 2$
- **gates**: $LD \times 2$
- **Output projection**: $2LD^2$

## H.5  Hyena-Y

- **Input projections**: $6LD^2$
- **Inner filter convolution**: $LDK \times 2$
- **gates**: $LD \times 2$
- **Output projection**: $2LD^2$

## H.6  Hyena-X (MLP)

- **Dense input projections**: $6LD^2r$
- **Featurizer**: $3LDK \times 2$
- **Gates**: $LD \times 2$
- **Dense output projections**: $2LD^2r$

### H.7 Mamba-2

- **Projections**: $8LD^2E$
- **Short convolution**: $6LDE$
- **Featurization**: $2LDE(1 + 2d_{\text{state}}) + 2LDE$
- **Associative scan**: $2LDEd_{\text{state}}$
- **Output layer**: $2LD^2E$

| Symbol | Description |
|:---:|:---|
| $L$ | Sequence length |
| $D$ | Hidden dimension |
| $H$ | Number of attention heads |
| $K$ | Kernel size for convolutions |
| $w$ | Window size for sliding window attention |
| $r$ | MLP expansion ratio |
| $E$ | Expansion factor in Mamba-2 |
| $d_{\text{state}}$ | State size in Mamba-2 |

Table H.1: Notation for FLOP calculation.

# I  Broader Impact, Limitations and Applications

**Broader Impact.** This work explores whether new architectural ideas can be materialized by editing pretrained diffusion transformers rather than training from scratch. By showing that architectural exploration is possible through grafting, our findings make model design more accessible and compute-efficient, enabling broader participation in generative modeling research. Grafting facilitates practical improvements—such as efficient operator replacement and structural reorganization—offering a simple tool for exploring generative model architectures. However, as more efficient methods lower the barrier to developing powerful models, they also increase risks of misuse, including the generation of misleading or harmful content. We encourage responsible use, transparency, and content safeguards when applying these techniques.

**Limitations.** This work primarily focuses on architectural editing of pretrained Diffusion Transformers (DiTs), specifically targeting self-attention and MLP operators. Other architectural components, such as normalization layers and activation functions, will be explored in future work. We note that these are latent diffusion models, and grafting components in their corresponding VAEs remains an area for future study. The PixArt-$\Sigma$ setup used synthetic data for grafting, which may propagate artifacts and biases into the grafted models. While this work focuses on architectural editing, it remains an open question whether architectures that perform well under grafting also perform well when trained from scratch. Finally, grafting requires access to a pretrained model.

**Applications and future work.** Grafting holds promise for diverse applications where model customization and efficiency are important. This includes adapting models from low-resolution to high-resolution settings, extending capabilities from short-form video understanding/generation to long-form [60, 61], or improving user experience in interactive applications like image editing where even modest gains (e.g., 10% speedup) are highly valued. We hope that our testbed, insights, and results will encourage the community to actively explore new architecture designs. Code and grafted models: grafting.stanford.edu.

# J   Compute details

We report the total compute used to run all experiments in this work, including self-grafting baselines, hybrid architectures, ablations, PixArt-$\Sigma$ adaptation, and depth-to-width restructuring. All experiments were conducted using 8×H100 GPUs. Table J.1 provides a detailed breakdown. We also include time spent on feature extraction and tuning.

Grafting enables efficient architectural exploration without pretraining. All experiments combined required approximately 6,050 H100 GPU hours, which remains significantly lower than the cost of training large diffusion transformers from scratch. Each hybrid experiment for DiT-XL/2 completes within 12–24 hours on 8×H100 GPUs (both Stage 1 and Stage 2).

| Experiment | H100 hours | # Experiments | Total H100 hours |
|---|---|---|---|
| Self-grafting Experiments (Tables 1 & 2) | 12 hrs × 8 | 4 | 384 |
| Hybrid Experiments (Table 4) | 12 hrs × 8 | 26 | 2,496 |
| Ablations | 24 hrs × 8 | 6 | 1,152 |
| PixArt-$\Sigma$ experiments | 63 hrs × 8 | 2 | 1,008 (Table 5) |
| Depth-to-width restructuring | 120 hrs × 8 | 1 | 960 (Table 6) |
| Other (feature extraction, tuning) | 50 hrs | — | 50 |
| **Total** | | | **6,050** |

Table J.1: Estimated compute usage across all experiments. Total GPU hours account for training, sampling, and evaluation. Each experiment was conducted using 8×H100 unless otherwise specified.

