# OpenReview forum: "Exploring Diffusion Transformer Designs via Grafting"
_NeurIPS.cc/2025/Conference — NeurIPS 2025 oral_

### Official Review · Reviewer_JnND · 2025-06-25

**Clarity:** 3
**Significance:** 2
**Originality:** 2
**Rating:** 4
**Confidence:** 4

**Summary:**

This paper considers to explore diffusion transformer via grafting. Specifically, it attempts to replace expensive operators (e.g., self-attention, MLPs) with efficient alternatives and restructures sequential computation into parallel.  Grafting includes two stages: Activation distillation and Lightweight fine-tuning. The authors conducts experiments on class-conditioned DiT and text-to-image PixArt, verifying the effectiveness of the proposed approach.

**Questions:**

1、Why MHA and MLP prefer different regression loss?

2、Actually, I may not agree with the author's opinion: 'We trade depth for width via grafting, achieving better quality than models of comparable depth.' Though the depth may be similar, e.g., DiT-B, TinyDiT-D14, and Grafting,  but the number of parameters is not. Is there a possibility that the better performance of Grafting is attributed to more model parameters?

3 See Weaknesses above.

**Ethical Concerns:**

["NO or VERY MINOR ethics concerns only"]

**Final Justification:**

No

**Limitations:**

The authors has discussed the limitations

**Quality:**

3

**Strengths And Weaknesses:**

Strengths

The exploration is interesting

well-organized and easy to follow

The results are reasonable

Weaknesses

Since obtaining a model with better performance is one of the most important goals in architecture design, this method lacks such a variant to illustrate the superiority of the method in this aspect. So far, it seems only be helpful in inference acceleration at a cost of performance. This method also may increase the parameters, for example in Table 6.

Considering the advantage of DiT in image and video generation,  conducting experiments only in image generation may be insufficient, it is highly recommended to employ grafting in video generation models. Similarly, can the proposed method enhance the model  performance in video generation.

For text-to-image generation, only using GenEval as the metric may not be sufficient.

---

> ### Author Rebuttal · Authors · 2025-07-31
>
> We thank the AC and the reviewer for their valuable time and feedback. In what follows, (1) we clarify the scope/goal of this work and address (2) requests for grafting experiments beyond image generation, (3) suggestions to include additional metrics beyond GenEval for T2I evaluation, and (4) clarifications on experimental details.
>
> $~$
> > W1: Since obtaining a model with better performance is one of the most important goals in architecture design, this method lacks such a variant to illustrate the superiority of the method in this aspect. So far, it seems only be helpful in inference acceleration at a cost of performance.
>
> We thank the reviewer for the thoughtful question. We agree that model quality is one of the important goals in architecture design. However, the goal of this work is not to outperform the base model, but to investigate whether diffusion transformer architectures can be edited post-training under small compute budgets, and to characterize the resulting quality–efficiency tradeoffs.
> Prior to this work, it was unknown whether such post-training architecture editing could preserve generation quality at all. Our results provide the first systematic study of post-training architectural edits on diffusion transformers (Tables 4, 5, and 6), mapping the quality–efficiency frontier in this setting.
>
> While some quality drop is expected when replacing expensive operators with cheaper alternatives, we quantify how much can be traded off, and demonstrate that architectural editing can preserve or closely match baseline generation quality.
>
> To summarize, the key outcomes of our work:
>
> - **Table 4 & Fig. 3 (Right):** We obtain post-training hybrid architectures that are within 0.5 FID of the DiT-XL/2 baseline, while grafting 50% of MHA operators (Hyena-SE/X/Y, SWA) and 100% of MLP operators (MLP r=3) with cheaper alternatives.
>
> - **Table 5:** On PixArt-Sigma, we obtain a 1.43× faster hybrid variant via operator replacement, with less than 2% GenEval score drop at 2048×2048 resolution.
>
> - **Table 6:** We restructure DiT-XL/2 by trading depth for width, producing a 2× shallower model (2× faster with fused implementation, bs=1) that achieves better FID (2.77) and outperforms pruning baselines and shallow-from-scratch trained models at depth=14. To our knowledge, this is the first attempt to convert sequential computation into parallel in DiTs post-training, enabling computation to be restructured.
>
>
> $~$
> > W2: Generalization beyond image generation
>
> We note that Generalization was a recurring theme raised by two other reviewers.
>
> - Reviewer r7Mh suggested experiments beyond “diffusion-based image generation” and “architectures beyond DiT”.
> - Reviewer VANM suggested “more open-source models with different architectures” with video generation as one possible experiment.
>
> **To address these concerns collectively, we evaluate grafting in the language modeling setting—a task, architecture, and modality distinct from diffusion-based image generation with DiTs.**
>
> **Setup:**
>
> - **Task:** Diffusion-based image generation → Autoregressive language modeling (next-token prediction)
> - **Architecture & Scale:** DiT-XL/2 (0.7B) → Qwen3 (4B)
> - **Operator:** MHA → Sliding Window Attention (window=256)
> - **Dataset**: Alpaca-cleaned (50k instruction-following examples)
> - **Evaluation:** PiQA, ARC-e, ARC-c, HellaSwag, Winogrande, MMLU following [F]
> - **Experiment details:** We apply our exact two-stage grafting procedure—activation distillation followed by lightweight fine-tuning. Stage 1 uses context length 1024; Stage 2 uses 8192.
>
> **Results:**
>
> Our grafted Qwen3-4B model achieves a **1.4x decode throughput** (at 8K context length with KV cache, Nvidia H100) with **less than 1% average performance drop**, demonstrating that grafting generalizes to new tasks, architectures, and modalities. We leave video generation to future work.
>
> | Model       | Stage        | Grafting Ratio | PiQA | ARC-e | ARC-c (acc. norm) | HellaSwag (acc. norm) | Winogrande | MMLU (5-shot) | Avg. | Decode throughput |
> | ----------- | ------------ | -------------- | ---- | ----- | ----------------- | --------------------- | ---------- | ------------- | ---- | ------- |
> | Qwen3       | Baseline     | N/A            | 74.9 | 80.5  | 54.0              | 68.5                  | 66.0       | 70.1          | 69.0 | \-      |
> | SWA (w=256) | Random Init. | 50%            | 64.3 | 56.5  | 29.7              | 37.9                  | 50.9       | 25.9          | 44.2 | 1.4x    |
> | SWA (w=256) | 1            | 50%            | 74.8 | 80.4  | 53.4              | 68.2                  | 65.5       | 66.9          | 68.2 | 1.4x    |
> | SWA (w=256) | 2            | 50%            | 75.5 | 80.3  | 52.3              | 69.6                  | 67.0       | 66.9          | 68.6 | 1.4x    |
>
>
> $~$
> > W3: For text-to-image generation, only using GenEval as the metric may not be sufficient.
>
> We thank the reviewer for this suggestion. While the reviewer did not specify a preferred metric, we note that GenEval is a widely adopted objective metric for evaluating prompt–image alignment in text-to-image generation [A,B]. Other common metrics such as IS and FID are less appropriate for high-resolution text-to-image tasks, as they rely on low-resolution ImageNet-1K features.
> To follow best practices, our paper already includes GenEval scores and qualitative samples (see Supplementary). To further strengthen the evaluation, we now include a small-scale human study, following recent work [C, D].
> We randomly sample 350 text–image pairs and ask three expert annotators to rate each image along two axes:
> - Prompt–image alignment: Does the image accurately reflect the text?
> - Realism: Does the image appear visually realistic?
>
> |          | Alignment | Realism | Speedup |
> | -------- | --------- | ------- | ------- |
> | Baseline | 0.87      | 0.85    | 1x      |
> | Hyena-X  | 0.82      | 0.79    | 1.43x   |
>
> These results show that the grafted variant (Hyena-X) maintains good quality across both axes. Together, we provide both objective (GenEval) and human (alignment/realism) evaluations of generation quality.
>
>
>
> $~$
> > Q1: Why MHA and MLP prefer different regression loss?
>
> This difference arises from benign overfitting. MLPs have 2× more parameters than MHA (10.6M vs. 5.3M), which makes them robust to outliers [E]. This behavior is also reflected in our validation loss curves (Supplementary D.3). For MHA, L1 regression consistently yields lower validation loss than L2, while the opposite holds for MLPs. We therefore adopt L1 for MHA and L2 for MLP. We will make this clear in the final version.
>
> $~$
> > Q2: Actually, I may not agree with the author's opinion: 'We trade depth for width via grafting, achieving better quality than models of comparable depth.' Though the depth may be similar, e.g., DiT-B, TinyDiT-D14, and Grafting, but the number of parameters is not. Is there a possibility that the better performance of Grafting is attributed to more model parameters?
>
> The reviewer is correct that the depth -> width restructured model has 6% more parameters (712M vs. 675M for DiT-XL/2) due to linear projections. However, **the goal of this case study is not to reduce parameter count, but to investigate whether deep, sequential transformer blocks can be restructured post-training into shallower, wider configurations that improve inference time while preserving quality.**
>
> In our restructuring, every two sequential DiT blocks are rewired post-training to run in parallel—reducing the model’s depth from 28 to 14 and achieving a 2× speedup (fused implementation, bs=1). Despite the slightly higher parameter count, the model achieves better quality (FID = 2.77) while benefiting from increased parallelism—demonstrating a favorable depth–width tradeoff.
> This experiment is not focused on parameter efficiency, but rather on post-training architectural restructuring—demonstrating, to our knowledge, **the first instance of block-level rewiring of pretrained DiTs**. For completeness, we include Table 6 below with speedup and FID metrics.
>
> | **Method**    | **Depth** | **A.R** | **Iters** | **IS ↑** | **FID ↓** | **sFID ↓** | **Prec. ↑** | **Recall ↑** | **Speedup ↑** | **Params ↓** |
> | ----------------------------------------- | --------: | ------: | --------: | -------: | --------: | ---------: | ----------: | -----------: | ------------: | -----------: |
> | DiT-L/2 |  24 |  42.7 | 1,000K | 196.26 | 3.73 |  4.62 | 0.82 | 0.54 | — |  458M |
> | U-ViT-L |  21 |  48.8 | 300K | 221.29 |  3.44 |  6.58 | 0.83 |  0.52 |  — |  287M |
> | DiT-B/2 | 12 |   64.0 | 1,000K |119.63 | 10.12 |  5.39 | 0.73 |0.55 |  — | 130M |
> | BK-SDM| 14 |  82.3 | 100K |141.18 | 7.43 |  6.09 | 0.75 | 0.55 |  2× |  340M |
> | TinyDiT-D14 |  14 |    82.3 | 500K |   198.85 | 3.92 | 5.69 | 0.78 |  0.58 | 2×| 340M |
> | TinyDiT-D14 w/ MKD |        14 |    82.3 | 500K |  234.50 | 2.86 | 4.75 | 0.82 |  0.55 |  2× | 340M |
> | DiT-XL/2 |  28 | 41.4 | 7,000K |   278.20 |  2.27 | 4.60 | 0.83 | 0.57 | 1× | 675M |
> | **Grafting** |14 | 164.6 | 100K |   231.91 | 3.12 |  4.71 | 0.82 |0.55 |2× | 712M |
> | **Grafting** |**14** | **164.6** |  **230K** |   **251.77** |  **2.77** | **4.87** |**0.82** |  **0.56** |       **2×** |         **712M** |
>
> **We hope our response addresses the reviewer's questions, and we’re happy to provide further details if helpful.**
>
> ====
>
> [A] Yao, Yuan, et al. "Diffusion Transformer-to-Mamba Distillation for High-Resolution Image Generation." arXiv (2025)
>
> [B] Tang, Bingda, et al. "Exploring the Deep Fusion of Large Language Models and Diffusion Transformers for Text-to-Image Synthesis." CVPR 2025
>
> [C] Cai, Shengqu, et al. "Diffusion self-distillation for zero-shot customized image generation." CVPR. 2025.
>
> [D] Chen, Junsong, et al. "Pixart-Sigma" ECCV, 2024.
>
> [E] Bartlett, Peter L., et al. "Benign overfitting in linear regression." PNAS 2020.
>
> [F] Zhang, Michael, et al. "LoLCATs: On Low-Rank Linearizing of Large Language Models." ICLR 2025

---

> > ### Author Response · Authors · 2025-08-07
> > **Seeking Feedback from Reviewer JnND**
> >
> > **Dear Reviewer JnND,**
> >
> > Thank you for your constructive and thoughtful review.
> >
> > With the author–reviewer discussion phase concluding in less than 48 hours, we would appreciate it if you could let us know whether our responses have addressed your concerns.
> >
> > **We have addressed all your comments and are happy to provide any further details.**
> >
> > Thank you for your time!
> >
> > Best,
> >
> > Authors

---

> > ### Comment · Reviewer_JnND · 2025-08-09
> >
> > Thank you for your detailed response and I decide to raise the score

---

> > > ### Author Response · Authors · 2025-08-09
> > > **Thank you Reviewer JnND for increasing your score**
> > >
> > > Dear Reviewer JnND,
> > >
> > > **Thank you for increasing your score.**
> > >
> > > Best,
> > >
> > > Authors.

---

### Official Review · Reviewer_w2ZL · 2025-06-30

**Clarity:** 2
**Significance:** 2
**Originality:** 2
**Rating:** 4
**Confidence:** 3

**Summary:**

This paper introduces grafting, a technique for architectural editing of pretrained diffusion transformers. it allows to replace expensive operators (such as self-attention or MLP layers) with more efficient alternatives in a two-stage process: (1) activation distillation and (2) lightweight fine-tuning. The method enables experimentation with hybrid architectures without full retraining, making it useful to explore new model designs with limited compute.

**Questions:**

Please see weaknesses. I also have the following questions and suggestions:

1. In scenarios where multiple types of operators are to be replaced, is there a recommended strategy? Should they be replaced all at once, or sequentially? Additionally, does the effectiveness of Stage 1 distillation degrade when replacing several components?

2. Are the hyperparameters from the original (base) model typically retained, or is it generally necessary to re-tune them after grafting?

Suggestion: For Table 4, I suggest that the authors consider adding the original FID values to plot (b), perhaps using dotted lines for visual reference. This would help contextualise the deltas and make the plot more informative.

**Ethical Concerns:**

["NO or VERY MINOR ethics concerns only"]

**Limitations:**

Please see weaknesses and questions.

**Paper Formatting Concerns:**

Following the NeurIPS formatting guidelines, please add the table captions above the tables.

**Quality:**

2

**Strengths And Weaknesses:**

*Strengths:*

1. The paper addresses a scalability challenge: architectural exploration in large generative models is expensive, which bottlenecks innovation.
2. The authors test a variety of edit types (operator swaps, depth/width changes) on large-scale diffusion transformers, with clear benchmarks on FID, and compute costs. The depth/width example of grafting shows that parallel computation (shallower, wider models) by preserving the quality of the model is possible.

*Weaknesses:*

1. While finetuning is lightweight, recovery of good performance may still require significant data/compute for certain tasks or very deep editing. Whether it always outperforms a fresh, small retrain is not clear for all cases.
2. Grafting is proposed as a method for exploring new architectures with limited compute. However, the authors have not trained their final proposed model from scratch for comparison against the grafted counterpart. As a result, it is unclear how much of the performance gain can be attributed to the distillation process itself. Additionally, unless I have misunderstood (and please feel free to clarify), it appears that the authors have not yet identified an architecture that outperforms their base model.

---

> ### Author Rebuttal · Authors · 2025-07-31
>
> We thank the AC and the reviewer for their helpful feedback. The reviewer appreciated our operator replacement and depth to width restructuring studies. The reviewer’s questions focus on: (1) whether lightweight fine-tuning remains modest in compute for deep edits and how this compares to small from-scratch retrains; (2) role of distillation and the feasibility of pretraining hybrids; (3) experimental details (replacement strategy, hyperparameters); and (4) presentation suggestions. Below, we address each point in detail.
>
> $~$
> > W1: While finetuning is lightweight, recovery of good performance may still require significant data/compute for certain tasks or very deep editing. Whether it always outperforms a fresh, small retrain is not clear for all cases.
>
> We agree that very deep edits can require more updates. Empirically, our **depth→width restructuring** reached **FID = 2.77 after 230k iterations** **(new result obtained during rebuttal)**, which is still tiny compared to the **~7M iterations** used to train the base model—i.e., even for deep edits we expect compute to remain **at least an order of magnitude lower than pretraining**. On “fresh small retrain”: the reviewer is correct to raise this. We note that **pretraining is out of scope** in our work as we focus on **post-training architectural editing** to study the resulting **quality–efficiency trade-offs**. Studying the relationship to from-scratch training is left to future work due to compute demands. Even pretraining a 64×64 ImageNet-1K DiT takes >12 days on 32×A100s [A] which is not feasible in academic compute budgets. We have noted this in our limitation section (Supp Sec. I).
>
>
> $~$
> > W2 (1): Grafting is proposed as a method for exploring new architectures with limited compute. However, the authors have not trained their final proposed model from scratch for comparison against the grafted counterpart. As a result, it is unclear how much of the performance gain can be attributed to the distillation process itself.
>
> The reviewer is correct—**distillation is important** for grafting, and **isolating its contribution** would require **training the hybrid architectures from scratch** for a head-to-head comparison. We note that pretraining is beyond the scope of this work, as we focus on **post-training architectural editing** of diffusion transformers. While an interesting direction, even training a 64x64 DiT takes more than 12 days on 32×A100s [A], making pretraining infeasible on an academic budget. That said, we agree this is an intellectually important question. To address it, we highlight independent evidence from language modeling showing that **hybrid architectures mixing attention with Hyena-style operators (including Hyena-SE) can be trained from scratch and remain competitive.**
>
> **Evidence: StripedHyena2: 7B-scale language models trained with Hyena-SE [B]:** StripedHyena2 [B] trains large-scale hybrid models from scratch using setups which mix attention and Hyena operators, including Hyena-SE. At the 7B scale, these models achieve **lower perplexity than transformer baselines** (2.83 vs. 3.09) after 400B tokens of pretraining—demonstrating that hybrid architectures remain expressive at scale.
>
> $~$
> > W2 (2) Additionally, unless I have misunderstood (and please feel free to clarify), it appears that the authors have not yet identified an architecture that outperforms their base model.
>
> We thank the reviewer for the thoughtful question. For clarity, **the goal of this work is not to outperform the base model but to investigate whether pretrained DiT architectures can be edited post-training under small compute budgets and to characterize the resulting quality–efficiency trade-offs.**
>
> Prior to this work, it was unknown whether such post-training architecture editing could preserve generation quality at all. Our results provide the first systematic study of post-training architectural edits on diffusion transformers (Tables 4, 5, and 6), mapping the quality–efficiency frontier in this setting.
>
> To summarize, the key outcomes of our work:
>
> - **Table 4 & Fig. 3 (Right):** We obtain post-training hybrid architectures that are within 0.5 FID of the DiT-XL/2 baseline, while grafting 50% of MHA operators (Hyena-SE/X/Y, SWA) and 100% of MLP operators (MLP r=3) with cheaper alternatives (<2% pretraining compute).
> - **Table 5:** On PixArt-Sigma, we obtain a 1.43× faster hybrid variant via operator replacement, with less than 2% GenEval score drop at 2048×2048 resolution.
> - **Table 6:** We restructure DiT-XL/2 by trading depth for width, producing a **2× shallower model** (2× faster with fused implementation, bs=1) that achieves **better FID (2.77)** and outperforms pruning baselines and shallow-from-scratch trained models at depth=14.
>
> Future work will explore pretraining and scaling of hybrid DiT architectures to improve model quality.
>
> $~$
> > Q1: In scenarios where multiple types of operators are to be replaced, is there a recommended strategy? Should they be replaced all at once, or sequentially? Additionally, does the effectiveness of Stage 1 distillation degrade when replacing several components?
>
> Replace **at once is recommended**; we observed only **minor improvements** over sequential replacements (we tried this—**FID improvements were < 0.07 for full MHA self-grafting**). Reviewer is correct, replacing more components can propagate error because stage 1 regression loss is not zero for every operator, but **Stage-2 (lightweight fine-tuning)** handles this. For example, under **full self-grafting** of MHA and MLP we achieve **near-baseline** performance (Table 2) after stage 2.
>
>
> $~$
> > Q2: Are the hyperparameters from the original (base) model typically retained, or is it generally necessary to re-tune them after grafting?
>
> Yes, base hyperparameters are reused. The only changes are (1) a simple linear warmup to the original base learning rate during Stage 2 (e.g., warming up to 1e‑4), and (2) a weight decay of 1e‑5 in stage 2. After warmup, the learning rate remains constant, following the original DiT recipe. Full hyperparameters for both Stage 1 and Stage 2 are provided in Supplement B.1.
>
> ####
> $~$
> > Suggestion: For Table 4, I suggest that the authors consider adding the original FID values to plot (b), perhaps using dotted lines for visual reference. This would help contextualise the deltas and make the plot more informative.
>
> Thank you for the suggestion. We will include this in the final version.
>
>
> **We hope our response addresses the reviewer's questions, and we’re happy to provide further details if helpful.**
>
> ====
>
> [A] Karras, Tero, et al. "Elucidating the design space of diffusion-based generative models." *NeurIPS 2022*
>
> [B] Ku, Jerome, et al. "Systems and algorithms for convolutional multi-hybrid language models at scale." *arXiv preprint arXiv:2503.01868* (2025).

---

> > ### Comment · Reviewer_w2ZL · 2025-08-05
> >
> > Thank you for the detailed response. I think that a comparison to distillation could be an interesting direction, though I understand that pretraining is beyond the scope of your current work. I will keep my positive score.

---

> > > ### Author Response · Authors · 2025-08-05
> > > **Author Response to Reviewer w2ZL**
> > >
> > > Dear Reviewer w2ZL,
> > >
> > > Thank you for maintaining a positive score. We agree that distillation is an interesting future direction. Thanks again for the feedback!

---

### Official Review · Reviewer_VANM · 2025-07-01

**Clarity:** 3
**Significance:** 3
**Originality:** 3
**Rating:** 5
**Confidence:** 4

**Summary:**

This article discusses the author's exploration of Diffusion Transformers (DiT) structure design using the Grafting technique. Grafting is employed for structural design research and demonstrates significant efficiency advantages compared to previous retraining-based approaches. Additionally, the author uses this technique to conduct a series of interesting explorations on diffusion model structures, such as replacing softmax attention with gated convolution, linear attention, and other notably advantageous alternative modules. Although this is an inspiring piece of work, there are still many confusing aspects and areas for improvement.

**Questions:**

Please see Weakness.

**Ethical Concerns:**

["NO or VERY MINOR ethics concerns only"]

**Final Justification:**

The authors have addressed most of my concerns in the rebuttal. I believe this work deserves to be accepted.

**Limitations:**

Yes.

**Paper Formatting Concerns:**

No formatting issues in this paper.

**Quality:**

3

**Strengths And Weaknesses:**

**Strengths:**
1. Grafting provides an efficient solution for exploring DiT structures and conducts explorations of certain value.
2. The article is well-written.
3. Experiments were conducted on DiT-XL/2 and PixArt-sigma, and the results are somewhat convincing.

**Weakness:**

Major Weakness:

1. Fine-tuning on pre-trained models leads to worse performance. The author employs various settings and uses the grafting method for fine-tuning, but almost all results in the tables show that the proposed methods underperform compared to the original pre-trained models. This diminishes the practical reference value of this work. While I understand the author's intention to explore interesting ideas, preserving generation quality is crucial for this task. The author should at least provide more analysis of the quality degradation.

2. Lack of comparison with existing work. Grafting is a technique that still requires minor fine-tuning. Since the acceleration effect is mentioned here, there are already numerous training-free acceleration techniques in the field that achieve DiT acceleration at almost no cost, such as the excellent works ToCa[1], TeaCache[2], PAB[3], and TaylorSeer[4]. When discussing acceleration, the author should compare with these methods or discuss their compatibility with such training-free approaches.


3. Concerns about generalization. The author's experiments were only conducted on DiT-XL/2 and PixArt-sigma, which do not represent the cutting-edge level of the current field, thereby reducing the persuasiveness of the author's claims. Migrating the proposed methods to more open-source models with different architectures, such as the text-to-image model FLUX[5] and video generation models like HunyuanVideo[6] and Wan2.1[7], and discussing them alongside existing acceleration methods from other directions (e.g., those mentioned in point 2) to demonstrate the generalization of the proposed approach is strongly encouraged.

[1] [ICLR2025]Accelerating Diffusion Transformers with Token-wise Feature Caching: https://arxiv.org/abs/2410.05317
[2] [CVPR2025]Timestep Embedding Tells: It's Time to Cache for Video Diffusion Model: https://arxiv.org/abs/2411.19108
[3] [ICLR2025] Real-Time Video Generation with Pyramid Attention Broadcast: https://arxiv.org/abs/2408.12588
[4] [ICCV2025] From Reusing to Forecasting: Accelerating Diffusion Models with TaylorSeers: https://arxiv.org/abs/2503.06923
[5] FLUX: https://github.com/black-forest-labs/flux
[6] HunyuanVideo: A Systematic Framework For Large Video Generative Models: https://arxiv.org/abs/2412.03603
[7] Wan2.1: Wan: Open and Advanced Large-Scale Video Generative Models: https://arxiv.org/abs/2503.20314

Minor Weakness:

The results in Table 6 should perhaps report the actual inference latency of the model. Additionally, the results in Table 6 are still not entirely satisfactory, and increasing the number of fine-tuning iterations might lead to more satisfactory outcomes.

---

> ### Author Rebuttal · Authors · 2025-07-31
>
> We thank the AC and reviewer for their valuable time and feedback. The reviewer raised several concerns: (1) generation quality gap and the practical relevance of grafting, (2) lack of comparison to caching methods, (3) limited generalization beyond DiT architectures and image generation, and (4) suggestions regarding latency reporting and improved results in Table 6. Below, we address all points in detail.
>
> $~$
> > W1: Generation quality gap and practical relevance
>
> The reviewer is correct, we agree that generation quality is essential for practical utility. However, the goal of this work is not to outperform the base model, but to investigate whether pretrained diffusion transformer architectures can be edited post-training under small compute budgets, and to characterize the resulting quality–efficiency tradeoffs.
> Prior to this work, it was unclear whether such post-training architecture editing could retain generation quality at all. Our results show that they can: across all experiments, the grafted architectures demonstrate good generation quality, despite significant architectural editing. For example:
>
> - **Table 4 & Fig. 3 (Right):** We obtain hybrid architectures that are within 0.5 FID of the DiT-XL/2 baseline, while grafting 50% of MHA layers (Hyena-SE/X/Y, SWA) and 100% of MLPs (MLP r=3).
>
> - **Table 5:** On PixArt-Σ, we obtain a 1.43× faster variant at 2048×2048 resolution with <2% GenEval score drop.
>
> - **Table 6:** We restructure DiT-XL/2 by trading depth for width, producing a 2× shallower model (2× faster with fused implementation, bs=1) that achieves better FID (2.77) and outperforms pruning baselines and shallow-from-scratch trained models at depth=14.
> While some quality drop is expected when replacing expensive operators with cheaper ones, our study quantifies these tradeoffs and demonstrates that high-quality generation is achievable when editing DiT architectures post-training.
>
> $~$
> > W2: Comparison to caching methods
>
> Thank you for raising this point. Caching methods like ToCa, TeaCache, PAB, and TaylorSeer accelerate inference by reusing intermediate activations at runtime. These techniques alter execution but leave the model’s architecture unchanged. In contrast, grafting edits the model’s architecture itself—replacing expensive operators/restructuring the computational graph post-training. **The two approaches are orthogonal and complementary. We will include a discussion of these works in the final version. Our submission also compares grafting with SOTA pruning methods in Section 6 (Table 6).**
>
>
> $~$
> > W3: Generalization beyond image generation
>
> We note that Generalization was a recurring theme raised by two other reviewers.
>
> - Reviewer r7Mh suggested experiments beyond “diffusion-based image generation” and “architectures beyond DiT”.
>
> - Reviewer JnND mentioned “experiments only in image generation may be insufficient” with video generation as one possible experiment.
>
> **To address these concerns collectively, we evaluate grafting in the language modeling setting—a task, architecture, and modality distinct from diffusion-based image generation with DiTs.**
>
> **Setup:**
>
> - **Task:** Diffusion-based image generation → Autoregressive language modeling (next-token prediction)
> - **Architecture & Scale:** DiT-XL/2 (0.7B) → Qwen3 (4B)
> - **Operator:** MHA → Sliding Window Attention (window=256)
> - **Dataset**: Alpaca-cleaned (50k instruction-following examples)
> - **Evaluation:** PiQA, ARC-e, ARC-c, HellaSwag, Winogrande, MMLU following [A]
> - **Experiment details:** We apply our exact two-stage grafting procedure—activation distillation followed by lightweight fine-tuning. Stage 1 uses context length 1024; Stage 2 uses 8192.
>
> **Results:**
>
> Our grafted Qwen3-4B model achieves a **1.4x decode throughput** (at 8K context length with KV cache, Nvidia H100) with **less than 1% average performance drop**, demonstrating that grafting generalizes to new tasks, architectures, and modalities.
>
> | Model       | Stage        | Grafting Ratio | PiQA | ARC-e | ARC-c (acc. norm) | HellaSwag (acc. norm) | Winogrande | MMLU (5-shot) | Avg. | Speedup |
> | ----------- | ------------ | -------------- | ---- | ----- | ----------------- | --------------------- | ---------- | ------------- | ---- | ------- |
> | Qwen3       | Baseline     | N/A            | 74.9 | 80.5  | 54.0              | 68.5                  | 66.0       | 70.1          | 69.0 | \-      |
> | SWA (w=256) | Random Init. | 50%            | 64.3 | 56.5  | 29.7              | 37.9                  | 50.9       | 25.9          | 44.2 | 1.4x    |
> | SWA (w=256) | 1            | 50%            | 74.8 | 80.4  | 53.4              | 68.2                  | 65.5       | 66.9          | 68.2 | 1.4x    |
> | SWA (w=256) | 2            | 50%            | 75.5 | 80.3  | 52.3              | 69.6                  | 67.0       | 66.9          | 68.6 | 1.4x    |
>
>
> $~$
> > Q1: Table 6 latency and improved FID results.
>
> We thank the reviewer for this suggestion. In the final version, we will add latency/ speedup relative to the baseline. As per reviewer’s suggestion, we have trained an improved variant that achieves a better FID (2.77) using 230k iterations. Below we include the updated table for your reference (Speedup measured using a fused implementation, bs=1).
>
> | **Method**    | **Depth** | **A.R** | **Iters** | **IS ↑** | **FID ↓** | **sFID ↓** | **Prec. ↑** | **Recall ↑** | **Speedup ↑** | **Params ↓** |
> | ----------------------------------------- | --------: | ------: | --------: | -------: | --------: | ---------: | ----------: | -----------: | ------------: | -----------: |
> | DiT-L/2 |  24 |  42.7 |    1,000K |   196.26 | 3.73 |  4.62 | 0.82 | 0.54 | — |  458M |
> | U-ViT-L |  21 |  48.8 |      300K |   221.29 |  3.44 |  6.58 | 0.83 |  0.52 |  — |  287M |
> | DiT-B/2 | 12 |   64.0 |    1,000K |   119.63 | 10.12 |  5.39 | 0.73 |0.55 |  — | 130M |
> | BK-SDM| 14 |  82.3 |      100K |   141.18 | 7.43 |  6.09 | 0.75 | 0.55 |       2× |         340M |
> | TinyDiT-D14 |  14 |    82.3 | 500K |   198.85 | 3.92 | 5.69 | 0.78 |         0.58 |       2×|         340M |
> | TinyDiT-D14 w/ MKD |        14 |    82.3 | 500K |  234.50 | 2.86 |       4.75 |        0.82 |         0.55 |       2× |         340M |
> | DiT-XL/2 |  28 | 41.4 | 7,000K |   278.20 |  2.27 | 4.60 | 0.83 |         0.57 |            1× |         675M |
> | **Grafting** |14 | 164.6 | 100K |   231.91 | 3.12 |  4.71 | 0.82 |         0.55 |       2× |         712M |
> | **Grafting** |**14** | **164.6** |  **230K** |   **251.77** |  **2.77** | **4.87** |**0.82** |  **0.56** |       **2×** |         **712M** |
>
>
> **We hope our response addresses the reviewer's questions, and we’re happy to provide further details if helpful. Thank you for the thoughtful feedback.**
>
> ===
>
> [A] Zhang, Michael, et al. "LoLCATs: On Low-Rank Linearizing of Large Language Models." ICLR 2025

---

> > ### Comment · Reviewer_VANM · 2025-08-05
> > **Response for Authors**
> >
> > Thank you for the authors' detailed clarifications and additional experiments during the rebuttal period. The majority of my concerns have been effectively addressed.
> >
> > I find it exciting that the proposed grafting technique achieves promising results on Large Language Models (LLMs). However, I still believe that demonstrating the technique's generalizability on generative models would significantly enhance the completeness and impact of this work. Showcasing the applicability of this grafting technique across a broader range of image and/or video generation models would make it a much more comprehensive contribution.

---

> > > ### Author Response · Authors · 2025-08-05
> > > **Author Response to Reviewer VANM**
> > >
> > > Dear Reviewer VANM,
> > >
> > > Thank you for the response. We’re glad most concerns are addressed and that you found the autoregressive LLM grafting results exciting. **We note that the goal of this work is to investigate post-training architecture editing**, and we hope to explore a broader range of image/ video generation models in future work. So far, we have studied:
> > > * **Generative modeling paradigms:** diffusion; autoregressive (rebuttal)
> > > * **Generative tasks:** class-conditional image generation; text-to-image generation; language modeling; depth→width architecture restructuring
> > > * **Architectures:** DiT-XL/2; PixArt-Σ; Qwen3-4B
> > > * **Sequence lengths:** 256, 1024, 8192, 16384
> > > * **Operator types:** MHA (local gated convolutions, local attention, linear attention); MLPs (variable expansion ratios, convolutional variants)
> > >
> > > We hope this addresses your remaining concern.
> > >
> > > Thanks again for the feedback!

---

> > > > ### Comment · Reviewer_VANM · 2025-08-06
> > > > **Response for Authors**
> > > >
> > > > Thank you for the response. While I appreciate the additional autoregressive-language-model experiments added during the rebuttal, I am particularly curious about how the proposed grafting technique influences efficiency in autoregressive generative models. Could the authors provide further results—such as FLOPs reduction, memory savings, or generation latency—obtained via grafting on such models? Including these metrics would significantly strengthen the paper’s completeness.

---

> ### Author Response · Authors · 2025-08-06
> **Follow-up Response to Reviewer VANM**
>
> Dear Reviewer VANM,
>
> Thank you for your follow-up.
>
> To address your question on how grafting affects efficiency in autoregressive LLMs, we report **decode throughput (tokens/s)** for the Qwen3-4B baseline and our grafted variant with 50% of MHA operators replaced by SWA (window=256). We observe:
>
> - **1.4× increase in decode throughput** over the baseline
> - **<1% drop in average performance** (68.6 vs. 69.0)
>
> | Model               | Grafting Ratio | Avg. Performance ↑| Decode Throughput ↑ (tokens/s) |
> |---------------------|----------------|---------------------|----------------------------------|
> | Qwen3-4B Baseline   | N/A             | 69.0                | 2600                        |
> | Grafted (SWA, w=256)| 50%            | 68.6                | 3695                       |
>
> Evaluated on a single NVIDIA H100 (KV cache, 8K context length, batch size ≈ 32).
>
> These results show that **grafting improves generation latency while preserving quality**.
>
> We hope this addresses your concern.
>
> Looking forward to your reply.
>
>
> Thanks!

---

> > ### Comment · Reviewer_VANM · 2025-08-07
> > **Response for Authors**
> >
> > Thank you for the clarifications and additional experiments provided in the rebuttal. The authors have addressed most of my concerns. I am very much looking forward to seeing this technique applied to video generation models in the future to further enhance the computational efficiency of such models. Meanwhile, I hope the authors can include more discussions on diffusion model acceleration in future versions, as this is a highly relevant and popular research area. I find the grafting technique very interesting and appreciate its broad applicability across various models. Based on these considerations, I am increasing my score to 5.

---

> > > ### Author Response · Authors · 2025-08-07
> > > **Thank you Reviewer VANM for increasing your score to 5**
> > >
> > > Dear Reviewer VANM,
> > >
> > > **Thank you for increasing your score to 5 .**
> > >
> > > We appreciate your thoughtful feedback and share your excitement about applying grafting to video generation models in future work. As per your suggestion, we will include a discussion on diffusion model acceleration in the final version.
> > >
> > > Thanks again for the feedback!
> > >
> > > Best,
> > >
> > > Authors

---

### Official Review · Reviewer_r7Mh · 2025-07-03

**Clarity:** 2
**Significance:** 2
**Originality:** 3
**Rating:** 4
**Confidence:** 4

**Summary:**

This paper introduces a method called Grafting, which is designed for editing a pretrained diffusion model to evaluate new network architectures with minimal computational cost. The approach consists of two stages: first, performing activation distillation for the new operator to be integrated; second, conducting lightweight finetuning after integration. The authors explore several architectural modifications and achieve comparable or even superior performance to the pretrained model by using this method, suggesting that new model designs can be efficiently explored by grafting pretrained model.

**Questions:**

- In Table 4(c), the interleaved strategy performs best, even outperforming the top-local strategy. Is there any explanation for this result? Does it suggest that the effectiveness of grafting is sensitive to the contextual operators surrounding the integrated operator?
- The paper uses synthetic data for two-stage training and observes some localized artifacts. Would such artifacts also appear when training on real data? Are there methods to solve this problem?
- Additional experiments on architectures beyond DiT or tasks beyond image generation would help demonstrate the broader applicability of the proposed method.
- It would be helpful to include from-scratch experiments on small-scale architectures to verify whether the relative performance of different designs is consistent with that observed under the grafting setting. This would further support the paper’s claim.

**Ethical Concerns:**

["NO or VERY MINOR ethics concerns only"]

**Final Justification:**

I appreciate the authors’ comprehensive rebuttal, which has satisfactorily addressed the majority of my concerns. Considering both the feedback provided by the other reviewers and the authors’ clarifications, I remain inclined to maintain my overall positive assessment of this submission.

**Limitations:**

Yes

**Paper Formatting Concerns:**

No.

**Quality:**

3

**Strengths And Weaknesses:**

**Strengths**

- The proposed method is simple and efficient, involving only two low-cost stages: activation distillation and lightweight finetuning. The methodology is clearly explained and with code and grafted models provided, making it easy to follow.

- The experiments in the paper are quite diverse and have good results. Most of the claims in the paper are supported by these experiments.
- The method has a wide range of potential applications, which is also supported by the experiments.

**Weaknesses**

- The method appears to be quite general and not inherently specific to diffusion models. But the experiments are conducted solely on diffusion-based image generation using the DiT architecture. Therefore, the generalizability of the approach remains unproven.
- One of the paper’s claims is that grafting can serve as an alternative method to exploring architectural effectiveness. However, it's unclear whether different operator might rely on their surrounding operators—since grafting requires freezing the context parameters. It also remains uncertain whether architectures that perform well under the grafting setting would yield similarly results when trained from scratch.

---

> ### Author Rebuttal · Authors · 2025-07-31
>
> We thank the AC and the reviewer for their constructive feedback. The reviewer highlighted the simplicity and efficiency of our approach and our experiments. The reviewer’s questions center on: (1) generalization of grafting beyond diffusion-based image generation using DiTs, (2) how operators interact with surrounding layers, (3) pretraining hybrid architectures, (4) potential artifacts from synthetic data, and (5) clarifications on experimental results. We respond in detail below:
>
> $~$
> > W1+Q3: Generalization beyond diffusion-based image generation using the DiT architecture.
>
> Thank you for your feedback. To address reviewer concerns, we evaluate grafting in an **autoregressive language modeling setting—a task, architecture, and modality distinct from diffusion-based image generation with DiTs.**
>
> **Setup:**
>
> - **Task:** Diffusion-based image generation → Autoregressive language modeling (next-token prediction)
> - **Architecture & Scale:** DiT-XL/2 (0.7B) → Qwen3 (4B)
> - **Operator:** MHA → Sliding Window Attention (window=256)
> - **Dataset**: Alpaca-cleaned (50k instruction-following examples)
> - **Evaluation:** PiQA, ARC-e, ARC-c, HellaSwag, Winogrande, MMLU following [E]
> - **Experiment details:** We apply our exact two-stage grafting procedure—activation distillation followed by lightweight fine-tuning. Stage 1 uses context length 1024; Stage 2 uses 8192.
>
> **Results:**
>
> Our grafted Qwen3-4B model achieves a **1.4x decode throughput** (at 8K context length with KV cache, Nvidia H100) with **less than 1% average performance drop**, demonstrating that grafting generalizes to new tasks, architectures, and modalities.
>
> | Model       | Stage        | Grafting Ratio | PiQA | ARC-e | ARC-c (acc. norm) | HellaSwag (acc. norm) | Winogrande | MMLU (5-shot) | Avg. | Speedup |
> | ----------- | ------------ | -------------- | ---- | ----- | ----------------- | --------------------- | ---------- | ------------- | ---- | ------- |
> | Qwen3       | Baseline     | N/A            | 74.9 | 80.5  | 54.0              | 68.5                  | 66.0       | 70.1          | 69.0 | \-      |
> | SWA (w=256) | Random Init. | 50%            | 64.3 | 56.5  | 29.7              | 37.9                  | 50.9       | 25.9          | 44.2 | 1.4x    |
> | SWA (w=256) | 1            | 50%            | 74.8 | 80.4  | 53.4              | 68.2                  | 65.5       | 66.9          | 68.2 | 1.4x    |
> | SWA (w=256) | 2            | 50%            | 75.5 | 80.3  | 52.3              | 69.6                  | 67.0       | 66.9          | 68.6 | 1.4x    |
>
>
> $~$
> > W2 (1) However, it's unclear whether different operator might rely on their surrounding operators—since grafting requires freezing the context parameters.
>
> The reviewer is correct: operator initialization relies on the surrounding operators. We freeze surrounding parameters only in Stage 1 (activation distillation). In Stage 2 (lightweight fine-tuning) we update all parameters end-to-end (see Fig. 1, Tables 4 and 6). For PixArt-Σ, we perform LoRA finetuning for memory efficiency (long sequence length = 16,384). We apply LoRA to self-attention (q/k/v/o), cross-attention (q, kv), and MLP (fc1/fc2), and update 1D causal convolutions in Hyena-X directly.
>
> $~$
> > W2 (2) + Q4: It also remains uncertain whether architectures that perform well under the grafting setting would yield similarly results when trained from scratch.
>
> Thank you for the thoughtful question. We note that pretraining is beyond the scope of this work, as we focus on **post-training architectural editing** of diffusion transformers. While an interesting direction, even training a 64x64 DiT takes more than 12 days on 32×A100s [A], making pretraining infeasible on an academic budget. That said, we agree this is an intellectually interesting question. To address it, we highlight independent evidence from language modeling showing that **hybrid architectures mixing attention with Hyena-style operators (including Hyena-SE) can be trained from scratch and remain competitive.**
>
> **Evidence: StripedHyena2: 7B-scale language models trained with Hyena-SE [B]:** StripedHyena2 [B] trains large-scale hybrid models from scratch using setups which mix attention and Hyena operators, including Hyena-SE. At the 7B scale, these models achieve **lower perplexity than transformer baselines** (2.83 vs. 3.09) after 400B tokens of pretraining—demonstrating that hybrid architectures are competitive at scale.
>
> $~$
> > Q1: In Table 4(c), the interleaved strategy performs best, even outperforming the top-local strategy. Is there any explanation for this result? Does it suggest that the effectiveness of grafting is sensitive to the contextual operators surrounding the integrated operator?
>
> Thank you for the thoughtful question. We believe the strong performance of the interleaved strategy stems from its ability to balance local and global receptive fields. In contrast, stacking multiple local operators (top-local) may restrict the model’s effective receptive field. Alternating with global operators helps retain broader context. Similar interleaving strategies have recently been shown effective in the pretraining of language models [B] and biological foundation models [C].
>
> $~$
> > Q2: The paper uses synthetic data for two-stage training and observes some localized artifacts. Would such artifacts also appear when training on real data? Are there methods to solve this problem?
>
> Thank you for the question. We do not expect such artifacts to appear when using real data. In Supplementary Fig. D.4 and D.4, we show that some low-quality synthetic samples can introduce artifacts during grafting. A practical way to mitigate this is to curate or filter synthetic data before use. Recent works [D] have explored automated data curation pipelines using VLMs for improving data quality.
>
>
> **We hope our response addresses the reviewer's questions, and we’re happy to provide further details if helpful. Thank you for the thoughtful feedback.**
>
> ####
>
> ####
>
> [A] Karras, Tero, et al. "Elucidating the design space of diffusion-based generative models." *NeurIPS 2022*
>
> [B] Ku, Jerome, et al. "Systems and algorithms for convolutional multi-hybrid language models at scale." *arXiv preprint arXiv:2503.01868* (2025).
>
> [C] Brixi, Garyk, et al. ‘Genome Modeling and Design across All Domains of Life with Evo 2’. *bioRxiv*, Cold Spring Harbor Laboratory, 2025.
>
> [D] Cai, Shengqu, et al. "Diffusion self-distillation for zero-shot customized image generation." CVPR 2025.
>
> [E] Zhang, Michael, et al. "LoLCATs: On Low-Rank Linearizing of Large Language Models." ICLR 2025

---

> > ### Comment · Reviewer_r7Mh · 2025-08-05
> >
> > I appreciate the authors' detailed response, which has successfully addressed most of my questions and concerns.
> >
> > The experimental results on autoregressive language models effectively demonstrate the generalizability of the proposed method. I recommend integrating these results as a section in the paper, as this would significantly clarify the contribution of this work.
> >
> > I acknowledge that pretraining falls outside the scope of this work and entails substantial computational overhead. But I also note that similar concerns were raised by Reviewer qERF and Reviewer w2ZL. This indicates that this topic represents a widely and valuable direction for future work.
> >
> > Overall, I maintain my positive assessment of this submission.

---

> > > ### Author Response · Authors · 2025-08-05
> > > **Author Response to Reviewer r7Mh**
> > >
> > > Dear Reviewer r7Mh,
> > >
> > > Thank you for the positive score. We're glad our response addressed most of your concerns. We will incorporate the autoregressive LLM grafting results into the final version, and we agree that pretraining hybrid architectures is an important future direction. Thanks again for the feedback!

---

### Official Review · Reviewer_qERF · 2025-07-05

**Clarity:** 3
**Significance:** 3
**Originality:** 4
**Rating:** 5
**Confidence:** 4

**Summary:**

- This work proposes grafting, a simple approach to edit pre-trained diffusion transformers (DiTs) to materialize new architectures. They do so by replacing expensive operators like Self Attention and MLP layers with efficient alternatives like Hyena, MAMBA etc.
- Authors perform an activation behavior and attention locality analysis on pre-trained DiT and study the impact of grafting on image quality using a DiT-XL/2 test bed.
- Authors successfully replace the multi-head attention with gated convolution and linear attention. Similarly they replace the MLPs with convolutional variants.
- Results show that many hybrid variants achieve competitive FID scores and on text-to-image models, grafting improves the speed of PixArt-$\Sigma$ by 43%.

**Questions:**

- Why was LoRA used for training PixArt-$\Sigma$? What is exactly being done here? Why do you need LoRA when you are finetuning all the replaced operators in the 2nd stage?
- What is the difference between Hyena SE and the two variants introduced in this work? Please add the differences between them in Fig. 3 Left. Why do you introduce causality in Hyena X and Y? (L205).
- If the emphasis of this work is to make operators efficient and speed up training/inference, have the authors considered edge applications? Does grafting only improve speed of operators or does it (with minor tweaks) also reduce the memory of the full network?

**Ethical Concerns:**

["NO or VERY MINOR ethics concerns only"]

**Final Justification:**

I thank the authors for their detailed rebuttal. They have answered my questions and concerns satisfactorily. Particularly my concern about general applicability of grafting has been answered properly. After reading through other reviews and author rebuttals, I vote to retain my score.

**Limitations:**

yes.

**Paper Formatting Concerns:**

No.

**Quality:**

4

**Strengths And Weaknesses:**

**Strengths**
- The activation behavior and locality analysis proposed by the authors are very interesting and can be useful diagnostic tools for understanding issues with pre-trained models. This kind of analysis can be very useful to the community. For example, from the locality analysis, one can replace the full attention in later parts of the model with local attention with little to no drop in quality.
- The writing is crisp and the experimental section is exhaustive. The experiments clearly corroborate the claims made in the paper.

**Weaknesses**
- The result that MHA grafting above 50% results in significant drop in performance tells us that the operators identified by grafting can never fully replace MHA limiting the applicability of grafting. I wonder if training from scratch with operators from grafting, can further limit the expressiveness of the model. While I understand that requesting to train a model from scratch is not within the scope of this work but understanding how the expressiveness of the model changes with the new operators is essential to understand the effectiveness of grafting. I believe this experiment is important because pre-trained weights are not always available for most models and for this work to be universally applicable, the dynamics of grafting when pre-trained from scratch is very important.

---

> ### Author Rebuttal · Authors · 2025-07-31
>
> We thank the AC and the reviewer for their valuable time and feedback. The reviewer appreciated our activation behavior and locality analysis, testbed and experiments. The reviewer raised concerns about training hybrid architectures from scratch, general questions about the experiment setup, and applications of grafting. Below, we address all comments.
>
> $~$
> > W1: Training from scratch with operators from grafting.
>
> Thank you for the thoughtful question. As noted by the reviewer, pretraining is beyond the scope of this work, as we focus on post-training architectural editing of diffusion transformers. While an interesting direction, even training a 64x64 DiT takes more than 12 days on 32×A100s [A], making pretraining infeasible on an academic budget. That said, we agree this is an intellectually interesting question. To address it, we highlight independent evidence from language modeling showing that hybrid architectures mixing attention with Hyena-style operators can be trained from scratch and remain competitive.
>
> **Evidence: StripedHyena2: 7B-scale language models trained with Hyena-SE [B]:**
> StripedHyena2 [B] trains large-scale hybrid models from scratch using setups which mix attention and Hyena operators, including Hyena-SE. At the 7B scale, these models achieve lower perplexity than transformer baselines (2.83 vs. 3.09) after 400B tokens of pretraining—demonstrating that hybrid architectures remain expressive at scale.
>
> $~$
> > Q1: Why was LoRA used for training PixArt-? What is exactly being done here? Why do you need LoRA when you are finetuning all the replaced operators in the 2nd stage?
>
> We used LoRA in our high-resolution PixArt-Σ experiments to reduce memory and compute overhead. As noted in lines 254–255, these experiments operate over long sequences (e.g., shape [B,16384,1152]), which makes standard finetuning expensive. In Stage 2 (lightweight finetuning), we apply LoRA to self-attention (q/k/v/o), cross-attention (q, kv), and MLP (fc1/fc2), and we update the 1D causal convolutions in Hyena-X directly (added to the optimizer; no LoRA, since these are 1D convs with a small parameter count). We will clarify this in the final version.
>
> $~$
> > Q2 (1) Difference between Hyena-SE, Hyena-X and Hyena-Y operators
>
> We summarize the differences below. The key distinctions lie in the use of convolutional featurizers and the presence or absence of the inner filter. As suggested by the reviewer, we will add Hyena-SE to Fig. 3 in the final version. We have also provided the operator equations in Supp. E.
>
> |          | Conv. Featurizers | Inner Filter               | FLOP equation |
> | -------- | ----------------- | -------------------------- | ------------- |
> | Hyena-SE | ✅	               | Short Explicit Convolution | Supp F.3      |
> | Hyena-X  | ✅	               | ❌                    | Supp F.4      |
> | Hyena-Y  | ❌           | Short Explicit Convolution | Supp F.5      |
>
> $~$
> > Q2 (2) Question reg. causality
>
> While our testbed (Table 3) already includes bidirectional operators (e.g., self-grafting, sliding window attention), we introduced causal Hyena operators to study causal inductive biases. Notably, causal variants perform competitively—Hyena-Y achieves a FID of 2.61, compared to 2.62 for Sliding Window Attention (SWA). If the reviewer prefers, we can include a bidirectional version of Hyena-X or Hyena-Y in the final version. This requires only a simple change: removing left-padding in the inner convolution and instead padding the input on the right.
>
> $~$
> > Q3 (2) Memory footprint of hybrid architectures
>
> The reviewer is correct in noting that efficient operators can reduce memory usage of the full model in addition to speed, depending on their hardware-aware implementation. We provide a reference: At a sequence length of 2048, peak memory is ~2.70 GB for a GQA Transformer++ baseline, versus ~2.55 GB for the corresponding Hyena-Y hybrid architecture (Δ −150 MB, ≈ −5.6%).
>
> $~$
> > Q3 (1) Edge applications
>
> The reviewer is correct—while this work studies grafting as a general-purpose architectural editing approach, replacing expensive components with cheaper alternatives naturally leads to faster inference/ lower memory usage, which are particularly valuable in edge scenarios.
>
> To further demonstrate broader applicability, we also applied grafting to large autoregressive language models. These experiments were conducted to address grafting’s generalization to other domains (different task/architectures) suggested by Reviewers r7Mh, VANM and JnND. We believe the results may be of interest here, especially in the context of real-time or edge deployment. By grafting a 36-layer **Qwen3 4 billion parameter language model with Sliding Window Attention operators (w=256, grafting ratio=50%)**, we obtain **1.4x decode throughput (at 8K context length, Nvidia H100 with KV cache) with less than 1% overall performance drop.**
>
> | Model       | Stage        | Grafting Ratio | PiQA | ARC-e | ARC-c (acc. norm) | HellaSwag (acc. norm) | Winogrande | MMLU (5-shot) | Avg. | Speedup |
> | ----------- | ------------ | -------------- | ---- | ----- | ----------------- | --------------------- | ---------- | ------------- | ---- | ------- |
> | Qwen3       | Baseline     | N/A            | 74.9 | 80.5  | 54.0              | 68.5                  | 66.0       | 70.1          | 69.0 | \-      |
> | SWA (w=256) | Random Init. | 50%            | 64.3 | 56.5  | 29.7              | 37.9                  | 50.9       | 25.9          | 44.2 | 1.4x    |
> | SWA (w=256) | 1            | 50%            | 74.8 | 80.4  | 53.4              | 68.2                  | 65.5       | 66.9          | 68.2 | 1.4x    |
> | SWA (w=256) | 2            | 50%            | 75.5 | 80.3  | 52.3              | 69.6                  | 67.0       | 66.9          | 68.6 | 1.4x    |
>
>
> **We hope our response addresses the reviewer's questions, and we’re happy to provide further details if helpful. Thank you for the thoughtful feedback.**
>
> ===
>
> [A] Karras, Tero, et al. "Elucidating the design space of diffusion-based generative models." NeurIPS 2022
>
> [B] Ku, Jerome, et al. "Systems and algorithms for convolutional multi-hybrid language models at scale." arXiv preprint arXiv:2503.01868 (2025).

---

> > ### Author Response · Authors · 2025-08-07
> > **Seeking Feedback from Reviewer qERF**
> >
> > **Dear Reviewer qERF,**
> >
> > Thank you for your constructive and thoughtful review.
> >
> > With the author–reviewer discussion phase concluding in less than 48 hours, we would appreciate it if you could let us know whether our responses have addressed your concerns.
> >
> > Thank you for your time!
> >
> > Best,
> >
> > Authors

---

### Author Response · Authors · 2025-08-04
**Request for Feedback on Rebuttal**

Dear Reviewers and Area Chair,

We sincerely appreciate your constructive and thoughtful reviews.

With the author-reviewer discussion phase concluding in less than three days, we would greatly appreciate it if you could let us know whether our responses have addressed your concerns.

**We have addressed all your comments, including additional experiments demonstrating the generalization of grafting to a new task, model architecture, and modality (autoregressive language modeling using Qwen3-4B). We’re happy to provide any further details.**

Thank you for your time.

Best,

Authors

---

### Author Response · Authors · 2025-08-09
**Author–Reviewer Discussion Summary**

Dear AC and reviewers,

Thank you for your thoughtful feedback and positive scores.

Reviewers raised three key concerns; our responses resolved them, as reflected in reviewer follow-ups.

1. **Generalization (r7Mh, VANM, JnND):** We applied grafting to a new generative task, architecture, and modality—autoregressive language modeling using Qwen3-4B—improving decode throughput by 1.4× at 8K context with <1% average performance drop, showing generalization beyond DiT/image diffusion.

2. **Pretraining hybrid architectures (qERF, r7Mh, w2ZL):** We clarified that the goal of this work is post-training architecture editing. We further cited independent evidence showing that hybrid architectures can be trained from scratch and are expressive at scale [A].

3. **Additional experiment details (VANM, qERF, w2ZL):** We provided efficiency metrics; clarified LoRA usage, Hyena-SE/X/Y operator details, and hyperparameters.

Thank you for your time and feedback!

Best,

Authors


===

[A] Ku, Jerome, et al. "Systems and algorithms for convolutional multi-hybrid language models at scale." arXiv preprint arXiv:2503.01868 (2025).

---

### Note · Authors · 2025-08-13

Dear AC and Reviewers,

Thank you for your thoughtful feedback and engagement throughout the review process.

**Author-reviewer discussion summary:** All reviewers acknowledged our rebuttal and the consensus is **highly positive, with final ratings of** **5** (qERF), **5** (VANM), **4** (r7Mh), **4** (w2ZL) and **4** (JnND — reviewer indicated they had raised their score but did not specify 4 or 5). Reviewers described our work as “**inspiring**” (VANM), “**exciting**” (VANM), “**very interesting**” (qERF, VANM), “**useful to the community**” (qERF), “**simple and efficient**” (r7Mh), “**wide range of potential applications/broad applicability**” (r7Mh, VANM), “**clear benchmarks**” (w2ZL), “**well-organized**” (JnND) and “**very much looking forward to seeing this technique applied to video generation models in the future**” (VANM). During the rebuttal, reviewers converged on three main concerns—generalization beyond diffusion/image tasks, pretraining hybrid architectures, and additional experimental details—which we addressed in full with new experiments, supporting evidence and clarifications.

For completeness, we summarize our thesis and key takeaways below:


**Thesis.** Investigating architectural design choices—operators and configurations—requires costly pretraining. We introduce **grafting**, a simple approach for editing pretrained diffusion transformers to materialize new architectures under small compute budgets. To our knowledge, this is the first systematic study of post-training architectural editing in Diffusion Transformers (DiTs).


**Key takeaways/results**

1. **Efficient hybrid architectures:** We materialize efficient hybrid architectures with good quality—**FID 2.38–2.64 vs. 2.27 for DiT-XL/2**—using **<2%** pretraining compute.
2. **Faster text-to-image generation:** On PixArt-Σ, grafting delivers a **1.43×** speedup at 2048×2048 with **<2%** drop in GenEval score.
3. **Depth → width architecture restructuring:** We restructure sequential computation into parallel post-training, yielding a **2× shallower** DiT-XL/2 model with competitive quality (e.g., **FID 2.77**), outperforming models of similar depth trained from scratch and pruning baselines.
4. **Autoregressive language model:** Applying grafting to **Qwen3-4B LLM** **improves decode throughput by 1.4×** at 8K context with **<1%** average performance drop.

PyTorch code and grafted models are included.

Thanks again for your time and feedback!


Best,

Authors

---

### Decision · Program_Chairs · 2025-09-17

**Decision:**

Accept (oral)

**Comment:**

The paper introduces grafting, a technique to edit trained models in terms of replacing a set of their operators with (potentially lower-weight) alternatives. Whereas the method likely has some generality to it, the paper focuses on diffusion transformers. Empirically, it shows that grafting can speed up various models with fairly low drops in performance. The strength of the paper certainly is a very innovative approach with a lot of potential for impact, especially as it is foreseeable to be applicable to other types of generation beyond what is investigated in the paper. The main weakness of the paper seems to be that the edited operators seem to come at a performance cost and don't seem to be able to truly replace the original. However, this cost seems more than acceptable and the innovativeness of the idea outshines the weakness by far.

Above sentiment is reflected by all reviewers, who, following the rebuttal, unanimously agree to accept the paper. They generally commend the paper for its well written nature and presentation, the novelty of the idea and the potential impact of grafting as a technique. Initial concerns all seemed to revolve around requests for further empirical evaluation and concerns over the observed drop in performance. However, following rebuttal discussions, all concerns were resolved and reviewers agree that the potential of the idea outweighs any remaining concerns. A particular concern on the experimentation, voiced by reviewer VANM, revolved around inclusion of further baselines and other method comparisons. In several rounds of discussions the authors have added a significant amount of further empirical evidence, broadening the scope of their analysis and further improving an already in-depth investigation. The reviewer has respectively decided to raise their score to a full accept recommendation and the AC views this discussion as also addressing some of the concerns of the other reviewers (who are already positive anyhow).

Overall, the paper presents an interesting and novel take on a very important problem. Given the well-written nature of the paper, the careful experimentation, and the high level of expected interest of a large community the AC recommends for the paper to be accepted with an oral presentation.